# Diverse Weight Averaging for Out-of-Distribution Generalization

**Alexandre Ramé[1],\*, Matthieu Kirchmeyer[1,2],\***
**Thibaud Rahier[2], Alain Rakotomamonjy[2,4], Patrick Gallinari[1,2], Matthieu Cord[1,3]**
[1]Sorbonne Université, CNRS, ISIR, F-75005 Paris, France   [2]Criteo AI Lab, Paris, France
[3]Valeo.ai, Paris, France   [4]Université de Rouen, LITIS, France
\*Equal contribution

## Abstract

Standard neural networks struggle to generalize under distribution shifts in computer vision. Fortunately, combining multiple networks can consistently improve out-of-distribution generalization. In particular, weight averaging (WA) strategies were shown to perform best on the competitive DomainBed benchmark; they directly average the weights of multiple networks despite their nonlinearities. In this paper, we propose Diverse Weight Averaging (DiWA), a new WA strategy whose main motivation is to increase the functional diversity across averaged models. To this end, DiWA averages weights obtained from several independent training runs: indeed, models obtained from different runs are more diverse than those collected along a single run thanks to differences in hyperparameters and training procedures. We motivate the need for diversity by a new bias-variance-covariance-locality decomposition of the expected error, exploiting similarities between WA and standard functional ensembling. Moreover, this decomposition highlights that WA succeeds when the variance term dominates, which we show occurs when the marginal distribution changes at test time. Experimentally, DiWA consistently improves the state of the art on DomainBed without inference overhead.

## 1   Introduction

Learning robust models that generalize well is critical for many real-world applications [1, 2]. Yet, the classical Empirical Risk Minimization (ERM) lacks robustness to distribution shifts [3, 4, 5]. To improve out-of-distribution (OOD) generalization in classification, several recent works proposed to train models simultaneously on multiple related but different domains [6]. Though theoretically appealing, domain-invariant approaches [7] either underperform [8, 9] or only slightly improve [10, 11] ERM on the reference DomainBed benchmark [12]. The state-of-the-art strategy on DomainBed is currently to average the weights obtained along a training trajectory [13]. [14] argues that this weight averaging (WA) succeeds in OOD because it finds solutions with flatter loss landscapes.

In this paper, we show the limitations of this flatness-based analysis and provide a new explanation for the success of WA in OOD. It is based on WA's similarity with ensembling [15], a well-known strategy to improve robustness [16, 17], that averages the predictions from various models. Based on [18], we present a bias-variance-covariance-locality decomposition of WA's expected error. It contains four terms: *first* the bias that we show increases under shift in label posterior distributions (i.e., correlation shift [19]); *second*, the variance that we show increases under shift in input marginal distributions (i.e., diversity shift [19]); *third*, the covariance that decreases when models are diverse; *finally*, a locality condition on the weights of averaged models.

Based on this analysis, we aim at obtaining diverse models whose weights are averageable with our Diverse Weight Averaging (DiWA) approach. In practice, DiWA averages in weights the models

obtained from independent training runs that share the same initialization. The motivation is that those models are more diverse than those obtained along a single run [20, 21]. Yet, averaging the weights of independently trained networks with batch normalization [22] and ReLU layers [23] may be counter-intuitive. Such averaging is efficient especially when models can be connected linearly in the weight space via a low loss path. Interestingly, this linear mode connectivity property [24] was empirically validated when the runs start from a shared pretrained initialization [25]. This insight is at the heart of DiWA but also of other recent works [26, 27, 28], as discussed in Section 6.

In summary, our main contributions are the following:

- We propose a new theoretical analysis of WA for OOD based on a bias-variance-covariance-locality decomposition of its expected error (Section 2). By relating correlation shift to its bias and diversity shift to its variance, we show that WA succeeds under diversity shift.
- We empirically tackle the covariance term by increasing the diversity across models averaged in weights. In our DiWA approach, we decorrelate their training procedures: in practice, these models are obtained from independent runs (Section 3). We then empirically validate that diversity improves OOD performance (Section 4) and show that DiWA is state of the art on all real-world datasets from the DomainBed benchmark [12] (Section 5).

## 2 Theoretical insights

Under the setting described in Section 2.1, we introduce WA in Section 2.2 and decompose its expected OOD error in Section 2.3. Then, we separately consider the four terms of this bias-variance-covariance-locality decomposition in Section 2.4. This theoretical analysis will allow us to better understand when WA succeeds, and most importantly, how to improve it empirically in Section 3.

### 2.1 Notations and problem definition

**Notations.** We denote $\mathcal{X}$ the input space of images, $\mathcal{Y}$ the label space and $\ell : \mathcal{Y}^2 \to \mathbb{R}_+$ a loss function. $S$ is the training (source) domain with distribution $p_S$, and $T$ is the test (target) domain with distribution $p_T$. For simplicity, we will indistinctly use the notations $p_S$ and $p_T$ to refer to the joint, posterior and marginal distributions of $(X, Y)$. We note $f_S, f_T : \mathcal{X} \to \mathcal{Y}$ the source and target labeling functions. We assume that there is no noise in the data: then $f_S$ is defined on $\mathcal{X}_S \triangleq \{x \in \mathcal{X}/p_S(x) > 0\}$ by $\forall (x, y) \sim p_S, f_S(x) = y$ and similarly $f_T$ is defined on $\mathcal{X}_T \triangleq \{x \in \mathcal{X}/p_T(x) > 0\}$ by $\forall (x, y) \sim p_T, f_T(x) = y$.

**Problem.** We consider a neural network (NN) $f(\cdot, \theta) : \mathcal{X} \to \mathcal{Y}$ made of a fixed architecture $f$ with weights $\theta$. We seek $\theta$ minimizing the target generalization error:

$$\mathcal{E}_T(\theta) = \mathbb{E}_{(x,y)\sim p_T}[\ell(f(x, \theta), y)]. \tag{1}$$

$f(\cdot, \theta)$ should approximate $f_T$ on $\mathcal{X}_T$. However, this is complex in the OOD setup because we only have data from domain $S$ in training, related yet different from $T$. The differences between $S$ and $T$ are due to distribution shifts (i.e., the fact that $p_S(X, Y) \neq p_T(X, Y)$) which are decomposed per [19] into **diversity shift** (a.k.a. covariate shift), when marginal distributions differ (i.e., $p_S(X) \neq p_T(X)$), and **correlation shift** (a.k.a. concept shift), when posterior distributions differ (i.e., $p_S(Y|X) \neq p_T(Y|X)$ and $f_S \neq f_T$). The weights are typically learned on a training dataset $d_S$ from $S$ (composed of $n_S$ i.i.d. samples from $p_S(X, Y)$) with a configuration $c$, which contains all other sources of randomness in learning (e.g., initialization, hyperparameters, training stochasticity, epochs, etc.). We call $l_S = \{d_S, c\}$ a learning procedure on domain $S$, and explicitly write $\theta(l_S)$ to refer to the weights obtained after stochastic minimization of $1/n_S \sum_{(x,y)\in d_S} \ell(f(x, \theta), y)$ w.r.t. $\theta$ under $l_S$.

### 2.2 Weight averaging for OOD and limitations of current analysis

**Weight averaging.** We study the benefits of combining $M$ individual member weights $\{\theta_m\}_{m=1}^M \triangleq \{\theta(l_S^{(m)})\}_{m=1}^M$ obtained from $M$ (potentially correlated) identically distributed (i.d.) learning procedures $L_S^M \triangleq \{l_S^{(m)}\}_{m=1}^M$. Under conditions discussed in Section 3.2, these $M$ weights can be averaged despite nonlinearities in the architecture $f$. Weight averaging (WA) [13], defined as:

$$f_{\text{WA}} \triangleq f(\cdot, \theta_{\text{WA}}), \text{ where } \theta_{\text{WA}} \triangleq \theta_{\text{WA}}(L_S^M) \triangleq 1/M \sum_{m=1}^M \theta_m, \tag{2}$$

is the state of the art [14, 29] on DomainBed [12] when the weights $\{\theta_m\}_{m=1}^M$ are sampled along a single training trajectory (a description we refine in Remark 1 from Appendix C.2).

**Limitations of the flatness-based analysis.** To explain this success, Cha *et al.* [14] argue that flat minima generalize better; indeed, WA flattens the loss landscape. Yet, as shown in Appendix B, this analysis does not fully explain WA's spectacular results on DomainBed. First, flatness does not act on distribution shifts thus the OOD error is uncontrolled with their upper bound (see Appendix B.1). Second, this analysis does not clarify why WA outperforms Sharpness-Aware Minimizer (SAM) [30] for OOD generalization, even though SAM directly optimizes flatness (see Appendix B.2). Finally, it does not justify why combining WA and SAM succeeds in IID [31] yet fails in OOD (see Appendix B.3). These observations motivate a new analysis of WA; we propose one below that better explains these results.

## 2.3 Bias-variance-covariance-locality decomposition

We now introduce our bias-variance-covariance-locality decomposition which extends the bias-variance decomposition [32] to WA. In the rest of this theoretical section, $\ell$ is the Mean Squared Error for simplicity: yet, our results may be extended to other losses as in [33]. In this case, the expected error of a model with weights $\theta(l_S)$ w.r.t. the learning procedure $l_S$ was decomposed in [32] into:

$$\mathbb{E}_{l_S}\mathcal{E}_T(\theta(l_S)) = \mathbb{E}_{(x,y)\sim p_T}[\text{bias}^2(x,y) + \text{var}(x)], \tag{BV}$$

where $\text{bias}(x,y), \text{var}(x)$ are the bias and variance of the considered model w.r.t. a sample $(x,y)$, defined later in Equation (BVCL). To decompose WA's error, we leverage the similarity (already highlighted in [13]) between WA and functional ensembling (ENS) [15, 34], a more traditional way to combine a collection of weights. More precisely, ENS averages the predictions, $f_{ENS} \triangleq f_{ENS}(\cdot, \{\theta_m\}_{m=1}^M) \triangleq 1/M \sum_{m=1}^M f(\cdot, \theta_m)$. Lemma 1 establishes that $f_{WA}$ is a first-order approximation of $f_{ENS}$ when $\{\theta_m\}_{m=1}^M$ are close in the weight space.

**Lemma 1** (WA and ENS. Proof in Appendix C.1. Adapted from [13, 28].)**.** *Given* $\{\theta_m\}_{m=1}^M$ *with learning procedures* $L_S^M \triangleq \{l_S^{(m)}\}_{m=1}^M$. *Denoting* $\Delta_{L_S^M} = \max_{m=1}^M \|\theta_m - \theta_{WA}\|_2$, $\forall(x,y) \in \mathcal{X} \times \mathcal{Y}$:

$$f_{WA}(x) = f_{ENS}(x) + O(\Delta_{L_S^M}^2) \text{ and } \ell(f_{WA}(x),y) = \ell(f_{ENS}(x),y) + O(\Delta_{L_S^M}^2).$$

This similarity is useful since Equation (BV) was extended into a bias-variance-covariance decomposition for ENS in [18, 35]. We can then derive the following decomposition of WA's expected test error. To take into account the $M$ averaged weights, the expectation is over the joint distribution describing the $M$ identically distributed (i.d.) learning procedures $L_S^M \triangleq \{l_S^{(m)}\}_{m=1}^M$.

**Proposition 1** (Bias-variance-covariance-locality decomposition of the expected generalization error of WA in OOD. Proof in Appendix C.2.)**.** *Denoting* $\bar{f}_S(x) = \mathbb{E}_{l_S}[f(x,\theta(l_S))]$, *under identically distributed learning procedures* $L_S^M \triangleq \{l_S^{(m)}\}_{m=1}^M$, *the expected generalization error on domain* $T$ *of* $\theta_{WA}(L_S^M) \triangleq \frac{1}{M}\sum_{m=1}^M \theta_m$ *over the joint distribution of* $L_S^M$ *is:*

$$\mathbb{E}_{L_S^M}\mathcal{E}_T(\theta_{WA}(L_S^M)) = \mathbb{E}_{(x,y)\sim p_T}\left[\text{bias}^2(x,y) + \frac{1}{M}\text{var}(x) + \frac{M-1}{M}\text{cov}(x)\right] + O(\bar{\Delta}^2),$$

$$\text{where bias}(x,y) = y - \bar{f}_S(x),$$

$$\text{and var}(x) = \mathbb{E}_{l_S}\left[\left(f(x,\theta(l_S)) - \bar{f}_S(x)\right)^2\right], \tag{BVCL}$$

$$\text{and cov}(x) = \mathbb{E}_{l_S,l_S'}\left[\left(f(x,\theta(l_S)) - \bar{f}_S(x)\right)\left(f(x,\theta(l_S'))) - \bar{f}_S(x)\right)\right],$$

$$\text{and } \bar{\Delta}^2 = \mathbb{E}_{L_S^M}\Delta_{L_S^M}^2 \text{ with } \Delta_{L_S^M} = \max_{m=1}^M\|\theta_m - \theta_{WA}\|_2.$$

cov *is the prediction covariance between two member models whose weights are averaged. The locality term* $\bar{\Delta}^2$ *is the expected squared maximum distance between weights and their average.*

Equation (BVCL) decomposes the OOD error of WA into four terms. The bias is the same as that of each of its i.d. members. WA's variance is split into the variance of each of its i.d. members divided by $M$ and a covariance term. The last locality term constrains the weights to ensure the validity of our approximation. In conclusion, combining $M$ models divides the variance by $M$ but introduces the covariance and locality terms which should be controlled along bias to guarantee low OOD error.

## 2.4 Analysis of the bias-variance-covariance-locality decomposition

We now analyze the four terms in Equation (BVCL). We show that bias dominates under correlation shift (Section 2.4.1) and variance dominates under diversity shift (Section 2.4.2). Then, we discuss a trade-off between covariance, reduced with diverse models (Section 2.4.3), and the locality term, reduced when weights are similar (Section 2.4.4). This analysis shows that *WA is effective against diversity shift when $M$ is large and when its members are diverse but close in the weight space*.

### 2.4.1 Bias and correlation shift (and support mismatch)

We relate OOD bias to correlation shift [19] under Assumption 1, where $\bar{f}_S(x) \triangleq \mathbb{E}_{l_S}[f(x, \theta(l_S))]$. As discussed in Appendix C.3.2, Assumption 1 is reasonable for a large NN trained on a large dataset representative of the source domain $S$. It is relaxed in Proposition 4 from Appendix C.3.

**Assumption 1** (Small IID bias). *$\exists \epsilon > 0$ small s.t. $\forall x \in \mathcal{X}_S, |f_S(x) - \bar{f}_S(x)| \leq \epsilon$.*

**Proposition 2** (OOD bias and correlation shift. Proof in Appendix C.3). *With a bounded difference between the labeling functions $f_T - f_S$ on $\mathcal{X}_T \cap \mathcal{X}_S$, under Assumption 1, the bias on domain $T$ is:*

$$\mathbb{E}_{(x,y) \sim p_T}[\text{bias}^2(x, y)] = \text{Correlation shift} + \text{Support mismatch} + O(\epsilon),$$

$$\text{where Correlation shift} = \int_{\mathcal{X}_T \cap \mathcal{X}_S} (f_T(x) - f_S(x))^2 p_T(x) dx, \tag{3}$$

$$\text{and Support mismatch} = \int_{\mathcal{X}_T \setminus \mathcal{X}_S} (f_T(x) - \bar{f}_S(x))^2 p_T(x) dx.$$

We analyze the first term by noting that $f_T(x) \triangleq \mathbb{E}_{p_T}[Y|X = x]$ and $f_S(x) \triangleq \mathbb{E}_{p_S}[Y|X = x]$, $\forall x \in \mathcal{X}_T \cap \mathcal{X}_S$. This expression confirms that our correlation shift term measures shifts in posterior distributions between source and target, as in [19]. It increases in presence of spurious correlations: e.g., on ColoredMNIST [8] where the color/label correlation is reversed at test time. The second term is caused by support mismatch between source and target. It was analyzed in [36] and shown irreducible in their "No free lunch for learning representations for DG". Yet, this term can be tackled if we transpose the analysis in the feature space rather than the input space. This motivates encoding the source and target domains into a shared latent space, e.g., by pretraining the encoder on a task with minimal domain-specific information as in [36].

This analysis explains why WA fails under correlation shift, as shown on ColoredMNIST in Appendix H. Indeed, combining different models does *not* reduce the bias. Section 2.4.2 explains that WA is however efficient against diversity shift.

### 2.4.2 Variance and diversity shift

Variance is known to be large in OOD [5] and to cause a phenomenon named underspecification, when models behave differently in OOD despite similar test IID accuracy. We now relate OOD variance to diversity shift [19] in a simplified setting. We fix the source dataset $d_S$ (with input support $X_{d_S}$), the target dataset $d_T$ (with input support $X_{d_T}$) and the network's initialization. We get a closed-form expression for the variance of $f$ over all other sources of randomness under Assumptions 2 and 3.

**Assumption 2** (Kernel regime). *$f$ is in the kernel regime [37, 38].*

This states that $f$ behaves as a Gaussian process (GP); it is reasonable if $f$ is a wide network [37, 39]. The corresponding kernel $K$ is the neural tangent kernel (NTK) [37] depending only on the initialization. GPs are useful because their variances have a closed-form expression (Appendix C.4.1). To simplify the expression of variance, we now make Assumption 3.

**Assumption 3** (Constant norm and low intra-sample similarity on $d_S$). *$\exists(\lambda_S, \epsilon)$ with $0 \leq \epsilon \ll \lambda_S$ such that $\forall x_S \in X_{d_S}, K(x_S, x_S) = \lambda_S$ and $\forall x'_S \neq x_S \in X_{d_S}, |K(x_S, x'_S)| \leq \epsilon$.*

This states that training samples have the same norm (following standard practice [39, 40, 41, 42]) and weakly interact [43, 44]. This assumption is further discussed and relaxed in Appendix C.4.2. We are now in a position to relate variance and diversity shift when $\epsilon \to 0$.

**Proposition 3** (OOD variance and diversity shift. Proof in Appendix C.4). *Given $f$ trained on source dataset $d_S$ (of size $n_S$) with NTK $K$, under Assumptions 2 and 3, the variance on dataset $d_T$ is:*

$$\mathbb{E}_{x_T \in X_{d_T}}[\text{var}(x_T)] = \frac{n_S}{2\lambda_S}MMD^2(X_{d_S}, X_{d_T}) + \lambda_T - \frac{n_S}{2\lambda_S}\beta_T + O(\epsilon), \qquad (4)$$

*where MMD is the empirical Maximum Mean Discrepancy in the RKHS of $K^2(x,y) = (K(x,y))^2$; $\lambda_T \triangleq \mathbb{E}_{x_T \in X_{d_T}} K(x_T, x_T)$ and $\beta_T \triangleq \mathbb{E}_{(x_T,x'_T) \in X^2_{d_T}, x_T \neq x'_T} K^2(x_T, x'_T)$ are the empirical mean similarities respectively measured between identical (w.r.t. $K$) and different (w.r.t. $K^2$) samples averaged over $X_{d_T}$.*

The MMD empirically estimates shifts in input marginals, i.e., between $p_S(X)$ and $p_T(X)$. Our expression of variance is thus similar to the diversity shift formula in [19]: MMD replaces the $L_1$ divergence used in [19]. The other terms, $\lambda_T$ and $\beta_T$, both involve internal dependencies on the target dataset $d_T$: they are constants w.r.t. $X_{d_T}$ and do not depend on distribution shifts. At fixed $d_T$ and under our assumptions, Equation (4) shows that variance on $d_T$ decreases when $X_{d_S}$ and $X_{d_T}$ are closer (for the MMD distance defined by the kernel $K^2$) and increases when they deviate. Intuitively, the further $X_{d_T}$ is from $X_{d_S}$, the less the model's predictions on $X_{d_T}$ are constrained after fitting $d_S$.

This analysis shows that WA reduces the impact of diversity shift as combining $M$ models divides the variance per $M$. This is a strong property achieved *without requiring data from the target domain*.

### 2.4.3   Covariance and diversity

The covariance term increases when the predictions of $\{f(\cdot, \theta_m)\}_{m=1}^M$ are correlated. In the worst case where all predictions are identical, covariance equals variance and WA is no longer beneficial. On the other hand, the lower the covariance, the greater the gain of WA over its members; this is derived by comparing Equations (BV) and (BVCL), as detailed in Appendix C.5. It motivates tackling covariance by encouraging members to make different predictions, thus to be functionally diverse. Diversity is a widely analyzed concept in the ensemble literature [15], for which numerous measures have been introduced [45, 46, 47]. In Section 3, we aim at decorrelating the learning procedures to increase members' diversity and reduce the covariance term.

### 2.4.4   Locality and linear mode connectivity

To ensure that WA approximates ENS, the last locality term $O(\bar{\Delta}^2)$ constrains the weights to be close. Yet, the covariance term analyzed in Section 2.4.3 is antagonistic, as it motivates functionally diverse models. Overall, to reduce WA's error in OOD, we thus seek a good trade-off between diversity and locality. In practice, we consider that the main goal of this locality term is to ensure that the weights are averageable despite the nonlinearities in the NN such that WA's error does not explode. This is why in Section 3, we empirically relax this locality constraint and simply require that the weights are linearly connectable in the loss landscape, as in the linear mode connectivity [24]. We empirically verify later in Figure 1 that the approximation $f_{\text{WA}} \approx f_{\text{ENS}}$ remains valid even in this case.

## 3   DiWA: Diverse Weight Averaging

### 3.1   Motivation: weight averaging from different runs for more diversity

**Limitations of previous WA approaches.**   Our analysis in Sections 2.4.1 and 2.4.2 showed that the bias and the variance terms are mostly fixed by the distribution shifts at hand. In contrast, the covariance term can be reduced by enforcing diversity across models (Section 2.4.3) obtained from learning procedures $\{l_S^{(m)}\}_{m=1}^M$. Yet, previous methods [14, 29] only average weights obtained along a single run. This corresponds to highly correlated procedures sharing the same initialization, hyperparameters, batch orders, data augmentations and noise, that only differ by the number of training steps. The models are thus mostly similar: this does not leverage the full potential of WA.

**DiWA.**   Our Diverse Weight Averaging approach seeks to reduce the OOD expected error in Equation (BVCL) by decreasing covariance across predictions: DiWA decorrelates the learning procedures $\{l_S^{(m)}\}_{m=1}^M$. Our weights are obtained from $M \gg 1$ different runs, with diverse learning procedures:

---

**Algorithm 1** DiWA Pseudo-code

---

**Require:** $\theta_0$ pretrained encoder and initialized classifier; $\{h_m\}_{m=1}^{H}$ hyperparameter configurations.

*Training:* $\forall m = 1$ to $H$, $\theta_m \triangleq \text{FineTune}(\theta_0, h_m)$

*Weight selection:*
   *Uniform:* $\mathcal{M} = \{1, \cdots, H\}$.
   *Restricted:* Rank $\{\theta_m\}_{m=1}^{H}$ by decreasing $\text{ValAcc}(\theta_m)$. $\mathcal{M} \leftarrow \emptyset$.
   **for** $m = 1$ to $H$ **do**
      **If** $\text{ValAcc}(\theta_{\mathcal{M} \cup \{m\}}) \geq \text{ValAcc}(\theta_{\mathcal{M}})$
      $\mathcal{M} \leftarrow \mathcal{M} \cup \{m\}$

*Inference:* with $f(\cdot, \theta_{\mathcal{M}})$, where $\theta_{\mathcal{M}} = \sum_{m \in \mathcal{M}} \theta_m / |\mathcal{M}|$.

---

these have different hyperparameters (learning rate, weight decay and dropout probability), batch orders, data augmentations (e.g., random crops, horizontal flipping, color jitter, grayscaling), stochastic noise and number of training steps. Thus, the corresponding models are more diverse on domain $T$ per [21] and reduce the impact of variance when $M$ is large. However, this may break the locality requirement analyzed in Section 2.4.4 if the weights are too distant. Empirically, we show that DiWA works under two conditions: shared initialization and mild hyperparameter ranges.

## 3.2 Approach: shared initialization, mild hyperparameter search and weight selection

**Shared initialization.** The shared initialization condition follows [25]: when models are fine-tuned from a shared pretrained model, their weights can be connected along a linear path where error remains low [24]. Following standard practice on DomainBed [12], our encoder is pretrained on ImageNet [48]; this pretraining is key as it controls the bias (by defining the feature support mismatch, see Section 2.4.1) and variance (by defining the kernel $K$, see Appendix C.4.4). Regarding the classifier initialization, we test two methods. The first is the random initialization, which may distort the features [49]. The second is Linear Probing (LP) [49]: it first learns the classifier (while freezing the encoder) to serve as a shared initialization. Then, LP fine-tunes the encoder and the classifier together in the $M$ subsequent runs; the locality term is smaller as weights remain closer (see [49]).

**Mild hyperparameter search.** As shown in Figure 5, extreme hyperparameter ranges lead to weights whose average may perform poorly. Indeed, weights obtained from extremely different hyperparameters may not be linearly connectable; they may belong to different regions of the loss landscape. In our experiments, we thus use the mild search space defined in Table 7, first introduced in SWAD [14]. These hyperparameter ranges induce diverse models that are averageable in weights.

**Weight selection.** The last step of our approach (summarized in Algorithm 1) is to choose which weights to average among those available. We explore two simple weight selection protocols, as in [28]. The first *uniform* equally averages all weights; it is practical but may underperform when some runs are detrimental. The second *restricted* (*greedy* in [28]) solves this drawback by restricting the number of selected weights: weights are ranked in decreasing order of validation accuracy and sequentially added only if they improve DiWA's validation accuracy.

In the following sections, we experimentally validate our theory. First, Section 4 confirms our findings on the OfficeHome dataset [50] where diversity shift dominates [19] (see Appendix E.2 for a similar analysis on PACS [51]). Then, Section 5 shows that DiWA is state of the art on DomainBed [12].

## 4 Empirical validation of our theoretical insights

We consider several collections of weights $\{\theta_m\}_{m=1}^{M}$ ($2 \leq M < 10$) trained on the "Clipart", "Product" and "Photo" domains from OfficeHome [50] with a shared random initialization and mild hyperparameter ranges. These weights are first indifferently sampled from a single run (every 50 batches) or from different runs. They are evaluated on "Art", the fourth domain from OfficeHome.

**WA vs. ENS.** Figure 1 validates Lemma 1 and that $f_{\text{WA}} \approx f_{\text{ENS}}$. More precisely, $f_{\text{WA}}$ slightly but consistently improves $f_{\text{ENS}}$: we discuss this in Appendix D. Moreover, a larger $M$ improves the

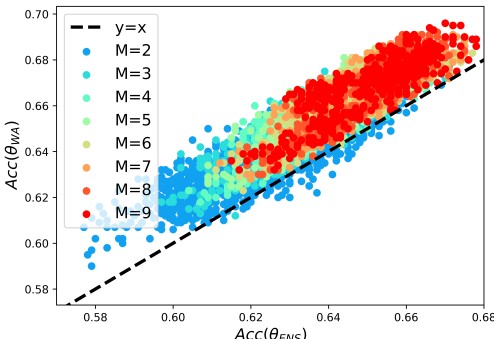

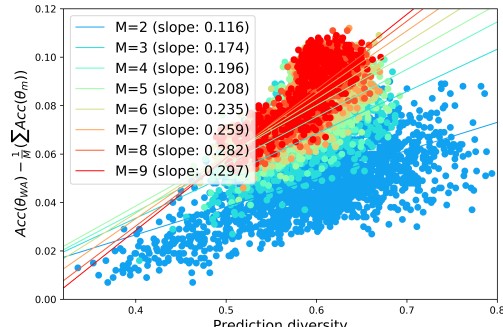

Figure 1: Each dot displays the accuracy (↑) of weight averaging (WA) vs. accuracy (↑) of prediction averaging (ENS) for $M$ models.

Figure 2: Each dot displays the accuracy (↑) gain of WA over its members vs. the prediction diversity [46] (↑) for $M$ models.

results; in accordance with Equation (BVCL), this motivates averaging as many weights as possible. In contrast, large $M$ is computationally impractical for ENS at test time, requiring $M$ forwards.

**Diversity and accuracy.** We validate in Figure 2 that $f_{\text{WA}}$ benefits from diversity. Here, we measure diversity with the ratio-error [46], i.e., the ratio $N_{\text{diff}}/N_{\text{simul}}$ between the number of different errors $N_{\text{diff}}$ and of simultaneous errors $N_{\text{simul}}$ in test for a pair in $\{f(\cdot, \theta_m)\}_{m=1}^M$. A higher average over the $\binom{M}{2}$ pairs means that members are less likely to err on the same inputs. Specifically, the gain of $\text{Acc}(\theta_{\text{WA}})$ over the mean individual accuracy $\frac{1}{M}\sum_{m=1}^M \text{Acc}(\theta_m)$ increases with diversity. Moreover, this phenomenon intensifies for larger $M$: the linear regression's slope (i.e., the accuracy gain per unit of diversity) increases with $M$. This is consistent with the $(M-1)/M$ factor of $\text{cov}(x)$ in Equation (BVCL), as further highlighted in Appendix E.1.2. Finally, in Appendix E.1.1, we show that the conclusion also holds with CKAC [47], another established diversity measure.

**Increasing diversity thus accuracy via different runs.** Now we investigate the difference between sampling the weights from a single run or from different runs. Figure 3 *first* shows that diversity increases when weights come from different runs. *Second*, in Figure 4, this is reflected on the accuracies in OOD. Here, we rank by validation accuracy the 60 weights obtained (1) from 60 different runs and (2) along 1 well-performing run. We then consider the WA of the top $M$ weights as $M$ increases from 1 to 60. Both have initially the same performance and improve with $M$; yet, WA of weights from different runs gradually outperforms the single-run WA. *Finally*, Figure 5 shows that this holds only for mild hyperparameter ranges and with a shared initialization. Otherwise, when hyperparameter distributions are extreme (as defined in Table 7) or when classifiers are not similarly initialized, DiWA may perform worse than its members due to a violation of the locality condition. These experiments confirm that *diversity is key as long as the weights remain averageable*.

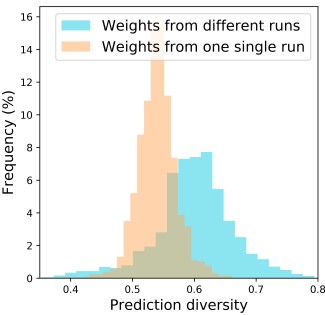

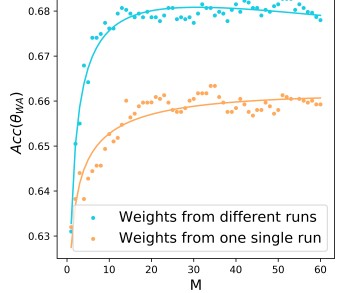

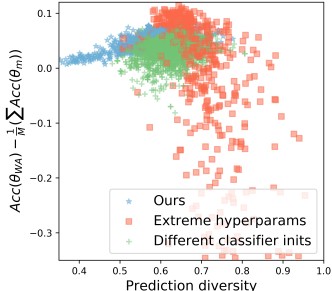

Figure 3: Frequencies of prediction diversities (↑) [46] across 2 weights obtained along a single run or from different runs.

Figure 4: WA accuracy (↑) as $M$ increases, when the $M$ weights are obtained along a single run or from different runs.

Figure 5: Each dot displays the accuracy (↑) gain of WA over its members vs. prediction diversity (↑) for $2 \le M < 10$ models.

# 5 Experimental results on the DomainBed benchmark

**Datasets.** We now present our evaluation on DomainBed [12]. By imposing the code, the training procedures and the ResNet50 [52] architecture, DomainBed is arguably the fairest benchmark for OOD generalization. It includes 5 multi-domain real-world datasets: PACS [51], VLCS [53], OfficeHome [50], TerraIncognita [54] and DomainNet [55]. [19] showed that *diversity shift dominates in these datasets*. Each domain is successively considered as the target $T$ while other domains are merged into the source $S$. The validation dataset is sampled from $S$, i.e., we follow DomainBed's training-domain model selection. The experimental setup is further described in Appendix G.1. Our code is available at `https://github.com/alexrame/diwa`.

**Baselines.** ERM is the standard Empirical Risk Minimization. Coral [10] is the best approach based on domain invariance. SWAD (Stochastic Weight Averaging Densely) [14] and MA (Moving Average) [29] average weights along one training trajectory but differ in their weight selection strategy. SWAD [14] is the current state of the art (SoTA) thanks to it "overfit-aware" strategy, yet at the cost of three additional hyperparameters (a patient parameter, an overfitting patient parameter and a tolerance rate) tuned per dataset. In contrast, MA [29] is easy to implement as it simply combines all checkpoints uniformly starting from batch 100 until the end of training. Finally, we report the scores obtained in [29] for the costly Deep Ensembles (DENS) [15] (with different initializations): we discuss other ensembling strategies in Appendix D.

**Our runs.** ERM and DiWA share the same training protocol in DomainBed: yet, instead of keeping only one run from the grid-search, DiWA leverages $M$ runs. In practice, we sample 20 configurations from the hyperparameter distributions detailed in Table 7 and report the mean and standard deviation across 3 data splits. For each run, we select the weights of the epoch with the highest validation accuracy. ERM and MA select the model with highest validation accuracy across the 20 runs, following standard practice on DomainBed. Ensembling (ENS) averages the predictions of all $M = 20$ models (with shared initialization). DiWA-restricted selects $1 \leq M \leq 20$ weights with Algorithm 1 while DiWA-uniform averages all $M = 20$ weights. DiWA$^\dagger$ averages uniformly the $M = 3 \times 20 = 60$ weights from all 3 data splits. DiWA$^\dagger$ benefits from larger $M$ (without additional inference cost) and from data diversity (see Appendix E.1.3). However, we cannot report standard deviations for DiWA$^\dagger$ for computational reasons. Moreover, DiWA$^\dagger$ cannot leverage the restricted weight selection, as the validation is not shared across all 60 weights that have different data splits.

## 5.1 Results on DomainBed

We report our **main results** in Table 1, detailed per domain in Appendix G.2. With a randomly initialized classifier, DiWA$^\dagger$-uniform is the best on PACS, VLCS and OfficeHome: DiWA-uniform is the second best on PACS and OfficeHome. On TerraIncognita and DomainNet, DiWA is penalized by some bad runs, filtered in DiWA-restricted which improves results on these datasets. Classifier initialization with linear probing (LP) [49] improves all methods on OfficeHome, TerraIncognita and DomainNet. On these datasets, DiWA$^\dagger$ increases MA by $1.3$, $0.5$ and $1.1$ points respectively. After averaging, DiWA$^\dagger$ with LP *establishes a new SoTA of* $68.0\%$, improving SWAD by $1.1$ points.

Table 1: **Accuracy** $(\%, \uparrow)$ **on DomainBed with ResNet50** (best in **bold** and second best underlined).

| | Algorithm | Weight selection | Init | PACS | VLCS | OfficeHome | TerraInc | DomainNet | Avg |
|---|---|---|---|---|---|---|---|---|---|
| | ERM | N/A | | $85.5 \pm 0.2$ | $77.5 \pm 0.4$ | $66.5 \pm 0.3$ | $46.1 \pm 1.8$ | $40.9 \pm 0.1$ | 63.3 |
| | Coral [10] | N/A | | $86.2 \pm 0.3$ | $78.8 \pm 0.6$ | $68.7 \pm 0.3$ | $47.6 \pm 1.0$ | $41.5 \pm 0.1$ | 64.6 |
| | SWAD [14] | Overfit-aware | Random | $88.1 \pm 0.1$ | $79.1 \pm 0.1$ | $70.6 \pm 0.2$ | $50.0 \pm 0.3$ | $46.5 \pm 0.1$ | 66.9 |
| | MA [29] | Uniform | | $87.5 \pm 0.2$ | $78.2 \pm 0.2$ | $70.6 \pm 0.1$ | $50.3 \pm 0.5$ | $46.0 \pm 0.1$ | 66.5 |
| | DENS [15, 29] | Uniform: $M = 6$ | | 87.6 | 78.5 | 70.8 | 49.2 | **47.7** | 66.8 |
| | ERM | N/A | | $85.5 \pm 0.5$ | $77.6 \pm 0.2$ | $67.4 \pm 0.6$ | $48.3 \pm 0.8$ | $44.1 \pm 0.1$ | 64.6 |
| | MA [29] | Uniform | | $87.9 \pm 0.1$ | $78.4 \pm 0.1$ | $70.3 \pm 0.1$ | $49.9 \pm 0.2$ | $46.4 \pm 0.1$ | 66.6 |
| | ENS | Uniform: $M = 20$ | Random | $88.0 \pm 0.1$ | $78.7 \pm 0.1$ | $70.5 \pm 0.1$ | $51.0 \pm 0.5$ | $47.4 \pm 0.2$ | 67.1 |
| | DiWA | Restricted: $M \leq 20$ | | $87.9 \pm 0.2$ | $\underline{79.2} \pm 0.1$ | $70.5 \pm 0.1$ | $50.5 \pm 0.5$ | $46.7 \pm 0.1$ | 67.0 |
| Our runs | DiWA | Uniform: $M = 20$ | | $88.8 \pm 0.4$ | $79.1 \pm 0.2$ | $71.0 \pm 0.1$ | $48.9 \pm 0.5$ | $46.1 \pm 0.1$ | 66.8 |
| | DiWA$^\dagger$ | Uniform: $M = 60$ | | **89.0** | **79.4** | 71.6 | 49.0 | 46.3 | 67.1 |
| | ERM | N/A | | $85.9 \pm 0.6$ | $78.1 \pm 0.5$ | $69.4 \pm 0.2$ | $50.4 \pm 1.8$ | $44.3 \pm 0.2$ | 65.6 |
| | MA [29] | Uniform | | $87.8 \pm 0.3$ | $78.5 \pm 0.4$ | $71.5 \pm 0.3$ | $51.4 \pm 0.6$ | $46.6 \pm 0.0$ | 67.1 |
| | ENS | Uniform: $M = 20$ | LP [49] | $88.1 \pm 0.3$ | $78.5 \pm 0.1$ | $71.7 \pm 0.1$ | $50.8 \pm 0.5$ | $47.0 \pm 0.2$ | 67.2 |
| | DiWA | Restricted: $M \leq 20$ | | $88.0 \pm 0.3$ | $78.5 \pm 0.1$ | $71.5 \pm 0.2$ | $\underline{51.6} \pm 0.9$ | **47.7** $\pm 0.1$ | 67.5 |
| | DiWA | Uniform: $M = 20$ | | $88.7 \pm 0.2$ | $78.4 \pm 0.2$ | $\underline{72.1} \pm 0.2$ | $51.4 \pm 0.6$ | $47.4 \pm 0.2$ | $\underline{67.6}$ |
| | DiWA$^\dagger$ | Uniform: $M = 60$ | | **89.0** | 78.6 | **72.8** | 51.9 | 47.7 | **68.0** |

**DiWA with different objectives.** So far we used ERM that does not leverage the domain information. Table 2 shows that DiWA-uniform benefits from averaging weights trained with Interdomain Mixup [56] and Coral [10]: accuracy gradually improves as we add more objectives. Indeed, as highlighted in Appendix E.1.3, DiWA benefits from the increased diversity brought by the various objectives. This suggests a new kind of linear connectivity across models trained with different objectives; the full analysis of this is left for future work.

Table 2: **Accuracy** $(\%, \uparrow)$ **on OfficeHome** domain "Art" with various objectives.

| Algorithm | No WA | MA | DiWA | DiWA$^{\dagger}$ |
|---|---|---|---|---|
| ERM | $62.9 \pm 1.3$ | $\underline{65.0} \pm 0.2$ | $67.3 \pm 0.2$ | 67.7 |
| Mixup | $\underline{63.1} \pm 0.7$ | $\mathbf{66.2} \pm 0.3$ | $67.8 \pm 0.6$ | 68.4 |
| Coral | $\mathbf{64.4} \pm 0.4$ | $64.4 \pm 0.4$ | $67.7 \pm 0.2$ | 68.2 |
| ERM/Mixup | N/A | N/A | $67.9 \pm 0.7$ | $\underline{68.9}$ |
| ERM/Coral | N/A | N/A | $\underline{68.1} \pm 0.3$ | 68.7 |
| ERM/Mixup/Coral | N/A | N/A | $\mathbf{68.4} \pm 0.4$ | **69.1** |

## 5.2 Limitations of DiWA

Despite this success, DiWA has some limitations. *First*, DiWA cannot benefit from additional diversity that would break the linear connectivity between weights — as discussed in Appendix D. *Second*, DiWA (like all WA approaches) can tackle diversity shift but not correlation shift: this property is explained for the first time in Section 2.4 and illustrated in Appendix H on ColoredMNIST.

## 6 Related work

**Generalization and ensemble.** To generalize under distribution shifts, invariant approaches [8, 9, 11, 10, 57, 58] try to detect the causal mechanism rather than memorize correlations: yet, they do not outperform ERM on various benchmarks [12, 19, 59]. In contrast, ensembling of deep networks [15, 60, 61] consistently increases robustness [16] and was successfully applied to domain generalization [29, 62, 63, 64, 65, 66]. As highlighted in [18] (whose analysis underlies our Equation (BVCL)), ensembling works due to the diversity among its members. This diversity comes primarily from the randomness of the learning procedure [15] and can be increased with different hyperparameters [67], data [68, 69, 70], augmentations [71, 72] or with regularizations [73, 65, 66, 74, 75].

**Weight averaging.** Recent works [13, 76, 77, 78] combine in weights (rather than in predictions) models collected along a single run. This was shown suboptimal in IID [17] but successful in OOD [14, 29]. Following the linear mode connectivity [24, 79] and the property that many independent models are connectable [80], a second group of works average weights with fewer constraints [26, 27, 28, 81, 82, 83]. To induce greater diversity, [84] used a high constant learning rate; [80] explicitly encouraged the weights to encompass more volume in the weight space; [83] minimized cosine similarity between weights; [85] used a tempered posterior. From a loss landscape perspective [20], these methods aimed at "explor[ing] the set of possible solutions instead of simply converging to a single point", as stated in [84]. The recent "Model soups" introduced by Wortsman *et al.* [28] is a WA algorithm similar to Algorithm 1; yet, the theoretical analysis and the goals of these two works are different. Theoretically, we explain why WA succeeds under diversity shift: the bias/correlation shift, variance/diversity shift and diversity-based findings are novel and are confirmed empirically. Regarding the motivation, our work aims at combining more diverse weights: it may be analyzed as a general framework to average weights obtained in various ways. In contrast, [28] challenges the standard model selection after a grid search. Regarding the task, [28] and our work complement each other: while [28] demonstrate robustness on several ImageNet variants with distribution shift, we improve the SoTA on the multi-domain DomainBed benchmark against other established OOD methods after a thorough and fair comparison. Thus, DiWA and [28] are theoretically complementary with different motivations and applied successfully for different tasks.

## 7 Conclusion

In this paper, we propose a new explanation for the success of WA in OOD by leveraging its ensembling nature. Our analysis is based on a new bias-variance-covariance-locality decomposition for WA, where we theoretically relate bias to correlation shift and variance to diversity shift. It also shows that diversity is key to improve generalization. This motivates our DiWA approach that averages in weights models trained independently. DiWA improves the state of the art on DomainBed, the reference benchmark for OOD generalization. Critically, DiWA has no additional inference cost — removing a key limitation of standard ensembling. Our work may encourage the community to further create diverse learning procedures and objectives — whose models may be averaged in weights.

## Acknowledgements

We would like to thank Jean-Yves Franceschi for his helpful comments and discussions on our paper. This work was granted access to the HPC resources of IDRIS under the allocation AD011011953 made by GENCI. We acknowledge the financial support by the French National Research Agency (ANR) in the chair VISA-DEEP (project number ANR-20-CHIA-0022-01) and the ANR projects DL4CLIM ANR-19-CHIA-0018-01, RAIMO ANR-20-CHIA-0021-01, OATMIL ANR-17-CE23-0012 and LEAUDS ANR-18-CE23-0020.

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
