# Appendices

This supplementary material complements the main paper. It is organized as follows:

## A Broader impact statement

We believe our paper can have several positive impacts. *First*, our theoretical analysis enables practitioners to know when averaging strategies succeed (under diversity shift, where variance dominates) or break down (under correlation shift, where bias dominates). This is key to understand when several models can be combined into a production system, or if the focus should be put on the training objective and/or the data. *Second*, it sets a new state of the art for OOD generalization under diversity shift without relying on a specific objective, architecture or task prior. It could be useful in medicine [1, 2] or to tackle fairness issues related to under-representation [57, 87, 88]. *Finally*, DIWA has no additional inference cost; in contrast, functional ensembling needs one forward per member. Thus, DiWA removes the carbon footprint overhead of ensembling strategies at test-time.

Yet, our paper may also have some negative impacts. *First*, it requires independent training of several models. It may motivate practitioners to learn even more networks and average them afterwards. Note that in Section 5, we restricted ourselves to combining only the runs obtained from the standard ERM grid search from DomainBed [12]. *Second*, our model is fully deep learning based with the corresponding risks, e.g., adversarial attacks and lack of interpretability. *Finally*, we do not control its possible use to surveillance or weapon systems.

## B Limitations of the flatness-based analysis in OOD

**Theorem 1** (Equation 21 from [14], simplified version of their Theorem 1). *Consider a set of $N$ covers $\{\Theta_k\}_{k=1}^N$ s.t. the parameter space $\Theta \subset \cup_k^N \Theta_k$ where $\mathrm{diam}(\Theta) \triangleq \sup_{\theta,\theta' \in \Theta} \|\theta - \theta'\|_2$, $N \triangleq \lceil (\mathrm{diam}(\Theta)/\gamma)^d \rceil$ and $d$ is the dimension of $\Theta$. Then, $\forall \theta \in \Theta$ with probability at least $1 - \delta$:*

$$
\begin{aligned}
\mathcal{E}_T(\theta) &\leq \frac{1}{2} \mathrm{Div}(p_S, p_T) + \mathcal{E}_S(\theta) \\
&\leq \frac{1}{2} \mathrm{Div}(p_S, p_T) + \mathcal{E}_{d_S}^\gamma(\theta) + \max_k \sqrt{\frac{(v_k[\ln(n_S/v_k) + 1] + \ln(N/\delta))}{2n_S}},
\end{aligned}
\tag{5}
$$

*where:*

- $\mathcal{E}_T(\theta) \triangleq \mathbb{E}_{(x,y) \sim p_T(X,Y)}[\ell(f_\theta(x); y)]$ *is the expected risk on the target domain,*

- $\mathrm{Div}(p_S, p_T) \triangleq 2\sup_A |p_S(A) - p_T(A)|$ *is a divergence between the source and target marginal distributions $p_S$ and $p_T$: it measures diversity shift.*

- $\mathcal{E}_S(\theta) \triangleq \mathbb{E}_{(x,y)\sim p_S(X,Y)}[\ell(f_\theta(x); y)]$ *is the expected risk on the source domain,*

- $\mathcal{E}_{d_S}^\gamma(\theta) \triangleq \max_{\|\Delta\|\leq\gamma} \mathcal{E}_{d_S}(\theta + \Delta)$ *(where $\mathcal{E}_{d_S}(\theta + \Delta) \triangleq \mathbb{E}_{(x,y)\in d_S}[\ell(f_{\theta+\Delta}(x); y)]$) is the robust empirical loss on source training dataset $d_S$ from $S$ of size $n_S$,*

- $v_k$ *is a VC dimension of each $\Theta_k$.*

Previous understanding of WA's success in OOD relied on this upper-bound, where $\mathcal{E}_{d_S}^\gamma(\theta)$ involves the solution's flatness. This is usually empirically analyzed by the trace of the Hessian [89, 90, 91]: indeed, with a second-order Taylor approximation around the local minima $\theta$ and $h$ the Hessian's maximum eigenvalue, $\mathcal{E}_{d_S}^\gamma(\theta) \approx \mathcal{E}_{d_S}(\theta) + h \times \gamma^2$.

In the following subsections, we show that this inequality does not fully explain the exceptional performance of WA on DomainBed [12]. Moreover, we illustrate that our bias-variance-covariance-locality addresses these limitations.

### B.1  Flatness does not act on distribution shifts

The flatness-based analysis is not specific to OOD. Indeed, the upper-bound in Equation (5) sums up two noninteracting terms: a domain divergence $\mathrm{Div}(p_S, p_T)$ that grows in OOD and $\mathcal{E}_{d_S}^\gamma(\theta)$ that measures the IID flatness. The flatness term can indeed be reduced empirically with WA: yet, it does not tackle the domain gap. In fact, Equation (5) states that additional flatness reduces the upper bound of the error similarly no matter the strength of the distribution shift, thus as well OOD than IID. In contrast, our analysis shows that variance (which grows with diversity shift, see Section 2.4.2) is tackled for large $M$: our error is controlled even under large diversity shift. This is consistent with our experiments in Table 1. Our analysis also explains why WA cannot tackle correlation shift (where bias dominates, see Appendix H), a limitation [14] does not illustrate.

### B.2  SAM leads to flatter minimas but worse OOD performance

The flatness-based analysis does not explain why WA outperforms other flatness-based methods in OOD. We consider Sharpness-Aware Minimizer (SAM) [30], another popular method to find flat minima based on minimax optimization: it minimizes the maximum loss around a neighborhood of the current weights $\theta$. In Figure 6, we compare the flatness (i.e., the Hessian trace computed with the package in [91]) and accuracy of ERM, MA [29] (a WA strategy) and SAM [30] when trained on the "Clipart", "Product" and "Photo" domains from OfficeHome [50]: they are tested OOD on the fourth domain "Art". Analyzing the second and the third rows of Figures 6a and 6b, we observe that SAM indeed finds flat minimas (at least comparable to MA), both in training (IID) and test (OOD). However, this is not reflected in the OOD accuracies in Figure 6c, where MA outperforms SAM. As reported in Table 3, similar experiments across more datasets lead to the same conclusions in [14]. In conclusion, flatness is not sufficient to explain why WA works so well in OOD, because SAM has similar flatness but worse OOD results. In contrast, we highlight in this paper that WA succeeds in OOD by reducing the impact of variance thanks to its similarity with prediction ensembling [15] (see Lemma 1), a privileged link that SAM does not benefit from.

Table 3: **Accuracy ($\uparrow$) on DomainBed for SWAD**, taken from Table 4 in [14]

|  | PACS | VLCS | OfficeHome | TerraInc | DomainNet | Avg. ($\Delta$) |
|---|---|---|---|---|---|---|
| ERM | $85.5 \pm 0.2$ | $77.5 \pm 0.4$ | $66.5 \pm 0.3$ | $46.1 \pm 1.8$ | $40.9 \pm 0.1$ | 63.3 |
| SWAD [14] + ERM | $\mathbf{88.1} \pm 0.1$ | $79.1 \pm 0.1$ | $\mathbf{70.6} \pm 0.2$ | $\mathbf{50.0} \pm 0.3$ | $\mathbf{46.5} \pm 0.1$ | $\mathbf{66.9}(+3.6)$ |
| SAM [30] | $85.8 \pm 0.2$ | $\mathbf{79.4} \pm 0.1$ | $69.6 \pm 0.1$ | $43.3 \pm 0.7$ | $44.3 \pm 0.0$ | 64.5 |
| SWAD [14] + SAM [30] | $87.1 \pm 0.2$ | $78.5 \pm 0.2$ | $69.9 \pm 0.1$ | $45.3 \pm 0.9$ | $\mathbf{46.5} \pm 0.1$ | $65.5(+1.0)$ |

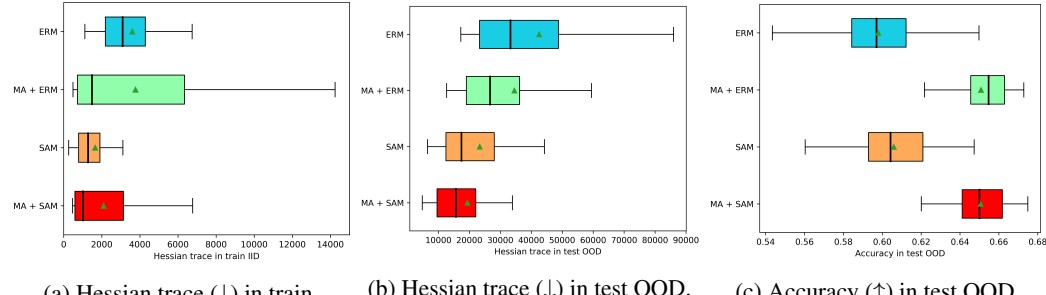

(a) Hessian trace (↓) in train.  (b) Hessian trace (↓) in test OOD.  (c) Accuracy (↑) in test OOD.

Figure 6: MA [29] (a WA strategy) and SAM [30] similarly improve flatness. When combined, they further improve flatness. Yet, MA outperforms SAM and beats MA + SAM in OOD accuracy on domain "Art" from OfficeHome.

Table 4: **Accuracy (↑) impact of including SAM** on domain "Art" from OfficeHome.

| Algorithm | Weight selection | ERM | SAM [30] |
|-----------|------------------|-----|----------|
| No DiWA | N/A | $62.9 \pm 1.3$ | $63.5 \pm 0.5$ |
| DiWA | Restricted: $M \leq 20$ | $66.7 \pm 0.1$ | $65.4 \pm 0.1$ |
| DiWA | Uniform: $M = 20$ | $67.3 \pm 0.3$ | $66.7 \pm 0.2$ |
| DiWA$^\dagger$ | Uniform: $M = 60$ | **67.7** | **67.4** |

### B.3 WA and SAM are not complementary in OOD when variance dominates

We investigate a similar inconsistency when combining these two flatness-based methods. As argued in [31], we confirm in Figures 6a and 6b that MA + SAM leads to flatter minimas than MA alone (i.e., with ERM) or SAM alone. Yet, MA does not benefit from SAM in Figure 6c. [14] showed an even stronger result in Table 3: SWAD + ERM performs better than SWAD + SAM. We recover similar findings in Table 4: DiWA performs worse when SAM is applied in each training run.

This behavior is not explained by Theorem 1, which states that more flatness should improve OOD generalization. Yet it is explained by our diversity-based analysis. Indeed, we observe in Figure 7 that the diversity across two checkpoints along a SAM trajectory is much lower than along a standard ERM trajectory (with SGD). We speculate that this is related to the recent empirical observation made in [92]: "the rank of the CLIP representation space is drastically reduced when training CLIP with SAM". Under diversity shift, variance dominates (see Equation (4)): in this setup, the gain in accuracy of models trained with SAM cannot compensate the decrease in diversity. This explains why WA and SAM are not complementary under diversity shift: in this case, variance is large.

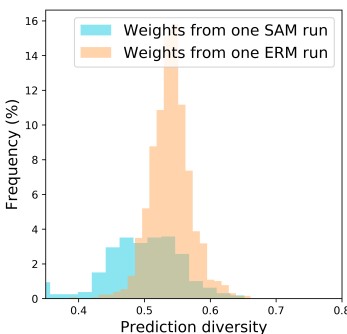

Figure 7: Prediction diversity in ratio-error [46] (↑) on domain "Art" from OfficeHome. Checkpoints along a SAM run are less diverse than along an ERM run.

## C Proof

### C.1 WA loss derivation

**Lemma** (1). *Given $\{\theta_m\}_{m=1}^M$ with learning procedures $L_S^M \triangleq \{l_S^{(m)}\}_{m=1}^M$. Denoting $\Delta_{L_S^M} = \max_{m=1}^M \|\theta_m - \theta_{WA}\|_2$, $\forall (x, y) \in \mathcal{X} \times \mathcal{Y}$:*

$$f_{WA}(x) = f_{ENS}(x) + O(\Delta_{L_S^M}^2) \text{ and } \ell(f_{WA}(x), y) = \ell(f_{ENS}(x), y) + O(\Delta_{L_S^M}^2).$$

*Proof.* This proof has two components:

- to establish the functional approximation, as [13], it performs Taylor expansion of the models' predictions at the first order.

- to establish the loss approximation, as [28], it performs Taylor expansion of the loss at the first order.

**Functional approximation**  With a Taylor expansion at the first order of the models' predictions w.r.t. parameters $\theta$:

$$f_{\theta_m} = f_{WA} + \nabla f_{WA}^{\mathsf{T}} \Delta_m + O\left(\|\Delta_m\|_2^2\right)$$

$$f_{ENS} - f_{WA} = \frac{1}{M} \sum_{m=1}^M \left(\nabla f_{WA}^{\mathsf{T}} \Delta_m + O\left(\|\Delta_m\|_2^2\right)\right)$$

Therefore, because $\sum_{m=1}^M \Delta_m = 0$,

$$f_{ENS} - f_{WA} = O\left(\Delta^2\right) \text{ where } \Delta = \max_{m=1}^M \|\Delta_m\|_2. \tag{6}$$

**Loss approximation**  With a Taylor expansion at the zeroth order of the loss w.r.t. its first input and injecting Equation (6):

$$\ell(f_{ENS}(x); y) = \ell(f_{WA}(x); y) + O(\|f_{ENS}(x) - f_{WA}(x)\|_2)$$

$$\ell(f_{ENS}(x); y) = \ell(f_{WA}(x); y) + O\left(\Delta^2\right).$$

$\square$

### C.2 Bias-variance-covariance-locality decomposition

**Remark 1.** *Our result in Proposition 1 is simplified by leveraging the fact that the learning procedures $L_S^M = \{l_S^{(m)}\}_{m=1}^M$ are identically distributed (i.d.). This assumption naturally holds for DiWA which selects weights from different runs with i.i.d. hyperparameters. It may be less obvious why it applies to MA [29] and SWAD [14]. It is even false if the weights $\{\theta(l_S^{(m)})\}_{m=1}^M$ are defined as being taken sequentially along a training trajectory, i.e., when $0 \leq i < j \leq M$ implies that $l_S^{(i)}$ has fewer training steps than $l_S^{(j)}$. We propose an alternative indexing strategy to respect the i.d. assumption. Given $M$ weights selected by the weight selection procedure, we draw without replacement the $M$ weights, i.e., $\theta(l_S^{(i)})$ refers to the $i^{th}$ sampled weights. With this procedure, all weights are i.d. as they are uniformly sampled. Critically, their WA are unchanged for the two definitions.*

**Proposition** (1). *Denoting $\bar{f}_S(x) = \mathbb{E}_{l_S}[f(x, \theta(l_S))]$, under identically distributed learning procedures $L_S^M \triangleq \{l_S^{(m)}\}_{m=1}^M$, the expected generalization error on domain $T$ of $\theta_{WA}(L_S^M) \triangleq$*

$\frac{1}{M}\sum_{m=1}^{M}\theta_m$ *over the joint distribution of* $L_S^M$ *is:*

$$\mathbb{E}_{L_S^M}\mathcal{E}_T(\theta_{WA}(L_S^M)) = \mathbb{E}_{(x,y)\sim p_T}\left[\text{bias}^2(x,y) + \frac{1}{M}\text{var}(x) + \frac{M-1}{M}\text{cov}(x)\right] + O(\bar{\Delta}^2),$$

$$\text{where bias}(x,y) = y - \bar{f}_S(x),$$

$$\text{and var}(x) = \mathbb{E}_{l_S}\left[\left(f(x,\theta(l_S)) - \bar{f}_S(x)\right)^2\right], \quad \text{(BVCL)}$$

$$\text{and cov}(x) = \mathbb{E}_{l_S,l'_S}\left[\left(f(x,\theta(l_S)) - \bar{f}_S(x)\right)\left(f(x,\theta(l'_S)) - \bar{f}_S(x)\right)\right],$$

$$\text{and } \bar{\Delta}^2 = \mathbb{E}_{L_S^M}\Delta_{L_S^M}^2 \text{ with } \Delta_{L_S^M} = \max_{m=1}^{M}\|\theta_m - \theta_{WA}\|_2.$$

cov *is the prediction covariance between two member models whose weights are averaged. The locality term* $\bar{\Delta}^2$ *is the expected squared maximum distance between weights and their average.*

*Proof.* This proof has two components:

- it follows the bias-variance-covariance decomposition from [18, 35] for functional ensembling. It is tailored to WA by assuming that learning procedures are identically distributed.

- it injects the obtained equation into Lemma 1 to obtain the Proposition 1 for WA.

**BVC for ensembling with identically distributed learning procedures** With $\bar{f}_S(x) = \mathbb{E}_{l_S}[f(x,\theta(l_S))]$, we recall the bias-variance decomposition [32] (Equation (BV)):

$$\mathbb{E}_{l_S}\mathcal{E}_T(\theta(l_S)) = \mathbb{E}_{(x,y)\sim p_T}\left[\text{bias}(x,y)^2 + \text{var}(x)\right],$$

$$\text{where bias}(x,y) = \text{Bias}\{f|(x,y)\} = y - \bar{f}_S(x),$$

$$\text{and var}(x) = \text{Var}\{f|x\} = \mathbb{E}_{l_S}\left[\left(f(x,\theta(l_S)) - \bar{f}_S(x)\right)^2\right].$$

Using $f_{\text{ENS}} \triangleq f_{\text{ENS}}(\cdot, \{\theta(l_S^{(m)})\}_{m=1}^M) \triangleq \frac{1}{M}\sum_{m=1}^{M}f(\cdot,\theta(l_S^{(m)}))$ in this decomposition yields,

$$\mathbb{E}_{L_S^M}\mathcal{E}_T(\{\theta(l_S^{(m)})\}_{m=1}^M) = \mathbb{E}_{x\sim p_T}\left[\text{Bias}\{f_{\text{ENS}} \mid (x,y)\}^2 + \text{Var}\{f_{\text{ENS}} \mid x\}\right]. \quad (7)$$

As $f_{\text{ENS}}$ depends on $L_S^M$, we extend the bias into:

$$\text{Bias}\{f_{\text{ENS}} \mid (x,y)\} = y - \mathbb{E}_{L_S^M}\left[\frac{1}{M}\sum_{m=1}^{M}f(x,\theta(l_S^{(m)}))\right] = y - \frac{1}{M}\sum_{m=1}^{M}\mathbb{E}_{l_S^{(m)}}\left[f(x,\theta(l_S^{(m)}))\right]$$

Under identically distributed $L_S^M \triangleq \{l_S^{(m)}\}_{m=1}^M$,

$$\frac{1}{M}\sum_{m=1}^{M}\mathbb{E}_{l_S^{(m)}}\left[y - f(x,\theta(l_S^{(m)}))\right] = \mathbb{E}_{l_S}[y - f(x,\theta(l_S))] = \text{Bias}\{f|(x,y)\}.$$

Thus the bias of ENS is the same as for a single member of the WA.

Regarding the variance:

$$\text{Var}\{f_{\text{ENS}} \mid x\} = \mathbb{E}_{L_S^M}\left[\left(\frac{1}{M}\sum_{m=1}^{M}f(x,\theta(l_S^{(m)})) - \mathbb{E}_{L_S^M}\left[\frac{1}{M}\sum_{m=1}^{M}f(x,\theta(l_S^{(m)}))\right]\right)^2\right].$$

Under identically distributed $L_S^M \triangleq \{l_S^{(m)}\}_{m=1}^M$,

$$\mathrm{Var}\{f_{\mathrm{ENS}} \mid x\} = \frac{1}{M^2} \sum_{m=1}^M \mathbb{E}_{l_S}\left[(f(x,\theta(l_S)) - \mathbb{E}_{l_S}[f(x,\theta(l_S))])^2\right] +$$

$$\frac{1}{M^2} \sum_m \sum_{m'\neq m} \mathbb{E}_{l_S,l'_S}\left[(f(x,\theta(l_S)) - \mathbb{E}_{l_S}[f(x,\theta(l_S))])(f(x,\theta(l'_S)) - \mathbb{E}_{l'_S}[f(x,\theta(l'_S))])\right]$$

$$= \frac{1}{M}\mathbb{E}_{l_S}\left[(f(x,\theta(l_S)) - \mathbb{E}_{l_S}[f(x,\theta(l_S))])^2\right] +$$

$$\frac{M-1}{M}\mathbb{E}_{l_S,l'_S}\left[(f(x,\theta(l_S)) - \mathbb{E}_{l_S}[f(x,\theta(l_S))])(f(x,\theta(l'_S)) - \mathbb{E}_{l'_S}[f(x,\theta(l'_S))])\right]$$

$$= \frac{1}{M}\mathrm{var}(x) + \left(1 - \frac{1}{M}\right)\mathrm{cov}(x).$$

The variance is split into the variance of a single member (divided by $M$) and a covariance term.

**Combination with Lemma 1** We recall that per Lemma 1,

$$\ell(f_{\mathrm{WA}}(x), y) = \ell(f_{\mathrm{ENS}}(x), y) + O(\Delta_{L_S^M}^2).$$

Then we have:

$$\mathcal{E}_T(\theta_{\mathrm{WA}}(L_S^M)) = \mathbb{E}_{(x,y)\sim p_T}[\ell(f_{\mathrm{WA}}(x), y)]$$

$$= \mathbb{E}_{(x,y)\sim p_T}[\ell(f_{\mathrm{ENS}}(x), y)] + O(\Delta_{L_S^M}^2) = \mathcal{E}_T(\{\theta(l_S^{(m)})\}_{m=1}^M) + O(\Delta_{L_S^M}^2),$$

$$\mathbb{E}_{L_S^M}\mathcal{E}_T(\theta_{\mathrm{WA}}(L_S^M)) = \mathbb{E}_{L_S^M}\mathcal{E}_T(\{\theta(l_S^{(m)})\}_{m=1}^M) + O(\mathbb{E}_{L_S^M}[\Delta_{L_S^M}^2]).$$

We eventually obtain the result:

$$\mathbb{E}_{L_S^M}\mathcal{E}_T(\theta_{\mathrm{WA}}(L_S^M)) = \mathbb{E}_{(x,y)\sim p_T}\left[\mathrm{bias}(x,y)^2 + \frac{1}{M}\mathrm{var}(x) + \frac{M-1}{M}\mathrm{cov}(x)\right] + O(\bar{\Delta}^2).$$

$\square$

## C.3 Bias, correlation shift and support mismatch

We first present in Appendix C.3.1 a decomposition of the OOD bias without any assumptions. We then justify in Appendix C.3.2 the simplifying Assumption 1 from Section 2.4.1.

### C.3.1 OOD bias

**Proposition 4** (OOD bias). *Denoting $\bar{f}_S(x) = \mathbb{E}_{l_S}[f(x,\theta(l_S))]$, the bias is:*

$$\mathbb{E}_{(x,y)\sim p_T}[\mathrm{bias}^2(x,y)] = \int_{\mathcal{X}_T\cap\mathcal{X}_S}(f_T(x) - f_S(x))^2 p_T(x)dx \qquad \textit{(Correlation shift)}$$

$$+ \int_{\mathcal{X}_T\cap\mathcal{X}_S}(f_S(x) - \bar{f}_S(x))^2 p_T(x)dx \qquad \textit{(Weighted IID bias)}$$

$$+ \int_{\mathcal{X}_T\cap\mathcal{X}_S}2(f_T(x) - f_S(x))(f_S(x) - \bar{f}_S(x))p_T(x)dx \qquad \textit{(Interaction IID bias and corr. shift)}$$

$$+ \int_{\mathcal{X}_T\setminus\mathcal{X}_S}(f_T(x) - \bar{f}_S(x))^2 p_T(x)dx. \qquad \textit{(Support mismatch)}$$

*Proof.* This proof is original and based on splitting the OOD bias in and out of $\mathcal{X}_S$:

$$\mathbb{E}_{(x,y)\sim p_T}[\mathrm{bias}^2(x,y)] = \mathbb{E}_{(x,y)\sim p_T}(y - \bar{f}_S(x))^2$$

$$= \int_{\mathcal{X}_T}(f_T(x) - \bar{f}_S(x))^2 p_T(x)dx$$

$$= \int_{\mathcal{X}_T\cap\mathcal{X}_S}(f_T(x) - \bar{f}_S(x))^2 p_T(x)dx + \int_{\mathcal{X}_T\setminus\mathcal{X}_S}(f_T(x) - \bar{f}_S(x))^2 p_T(x)dx.$$

To decompose the first term, we write $\forall x \in \mathcal{X}_S, -\bar{f}_S(x) = -f_S(x) + \big(f_S(x) - \bar{f}_S(x)\big)$.

$$\int_{\mathcal{X}_T \cap \mathcal{X}_S} \big(f_T(x) - \bar{f}_S(x)\big)^2 p_T(x) dx = \int_{\mathcal{X}_T \cap \mathcal{X}_S} \big((f_T(x) - f_S(x)) + \big(f_S(x) - \bar{f}_S(x)\big)\big)^2 p_T(x) dx$$

$$= \int_{\mathcal{X}_T \cap \mathcal{X}_S} (f_T(x) - f_S(x))^2 p_T(x) dx + \int_{\mathcal{X}_T \cap \mathcal{X}_S} \big(f_S(x) - \bar{f}_S(x)\big)^2 p_T(x) dx$$

$$+ \int_{\mathcal{X}_T \cap \mathcal{X}_S} 2(f_T(x) - f_S(x))\big(f_S(x) - \bar{f}_S(x)\big) p_T(x) dx.$$

$\square$

The four terms can be qualitatively analyzed:

- The first term measures differences between train and test labelling function. By rewriting $\forall x \in \mathcal{X}_T \cap \mathcal{X}_S, f_T(x) \triangleq \mathbb{E}_{p_T}[Y|X = x]$ and $f_S(x) \triangleq \mathbb{E}_{p_S}[Y|X = x]$, this term measures whether conditional distributions differ. This recovers a similar expression to the correlation shift formula from [19].

- The second term is exactly the IID bias, but weighted by the marginal distribution $p_T(X)$.

- The third term $\int_{\mathcal{X}_T \cap \mathcal{X}_S} 2(f_T(x) - f_S(x))\big(f_S(x) - \bar{f}_S(x)\big) p_T(x) dx$ measures to what extent the IID bias compensates the correlation shift. It can be negative if (by chance) the IID bias goes in opposite direction to the correlation shift.

- The last term measures support mismatch between test and train marginal distributions. It lead to the "No free lunch for learning representations for DG" in [36]. The error is irreducible because "outside of the source domain, the label distribution is unconstrained": "for any domain which gives some probability mass on an example that has not been seen during training, then all [...] labels for that example" are possible.

### C.3.2  Discussion of the small IID bias Assumption 1

Assumption 1 states that $\exists \epsilon > 0$ small s.t. $\forall x \in \mathcal{X}_S, |f_S(x) - \bar{f}_S(x)| \leq \epsilon$ where $\bar{f}_S(x) = \mathbb{E}_{l_S}[f(x, \theta(l_S))]$. $\bar{f}_S$ is the expectation over the possible learning procedures $l_S = \{d_S, c\}$. Thus Assumption 1 involves:

- the network architecture $f$ which should be able to fit a given dataset $d_S$. This is realistic when the network is sufficiently parameterized, i.e., when the number of weights $|\theta|$ is large.

- the expected datasets $d_S$ which should be representative enough of the underlying domain $S$; in particular the dataset size $n_S$ should be large.

- the sampled configurations $c$ which should be well chosen: the network should be trained for enough steps, with an adequate learning rate ...

For DiWA, this is realistic as it selects the weights with the highest training validation accuracy from each run. For SWAD [14], this is also realistic thanks to their overfit-aware weight selection strategy. In contrast, this assumption may not perfectlty hold for MA [29], which averages weights starting from batch 100 until the end of training: indeed, 100 batches are not enough to fit the training dataset.

### C.3.3  OOD bias when small IID bias

We now develop our equality under Assumption 1.

**Proposition** (2. OOD bias when small IID bias). *With a bounded difference between the labeling functions $f_T - f_S$ on $\mathcal{X}_T \cap \mathcal{X}_S$, under Assumption 1, the bias on domain T is:*

$$\mathbb{E}_{(x,y) \sim p_T}[\text{bias}^2(x, y)] = \text{Correlation shift} + \text{Support mismatch} + O(\epsilon),$$

$$\text{where Correlation shift} = \int_{\mathcal{X}_T \cap \mathcal{X}_S} (f_T(x) - f_S(x))^2 p_T(x) dx, \quad (3)$$

$$\text{and Support mismatch} = \int_{\mathcal{X}_T \setminus \mathcal{X}_S} \big(f_T(x) - \bar{f}_S(x)\big)^2 p_T(x) dx.$$

*Proof.* We simplify the second and third terms from Proposition 4 under Assumption 1.

**The second term** is $\int_{\mathcal{X}_T \cap \mathcal{X}_S} \big(f_S(x) - \bar{f}_S(x)\big)^2 p_T(x)dx$. Under Assumption 1, $|f_S(x) - \bar{f}_S(x)| \leq \epsilon$. Thus the second term is $O(\epsilon^2)$.

**The third term** is $\int_{\mathcal{X}_T \cap \mathcal{X}_S} 2(f_T(x) - f_S(x))\big(f_S(x) - \bar{f}_S(x)\big)p_T(x)dx$. As $f_T - f_S$ is bounded on $\mathcal{X}_S \cap \mathcal{X}_T$, $\exists K \geq 0$ such that $\forall x \in \mathcal{X}_S$,

$$|(f_T(x) - f_S(x))\big(f_S(x) - \bar{f}_S(x)\big)p_T(x)| \leq K|f_S(x) - \bar{f}_S(x)|p_T(x) = O(\epsilon)p_T(x).$$

Thus the third term is $O(\epsilon)$.

Finally, note that we cannot say anything about $\bar{f}_S(x)$ when $x \in \mathcal{X}_T \setminus \mathcal{X}_S$. $\qquad \square$

To prove the previous equality, we needed a bounded difference between labeling functions $f_T - f_S$ on $\mathcal{X}_T \cap \mathcal{X}_S$. We relax this bounded assumption to obtain an inequality in the following Proposition 5.

**Proposition 5** (OOD bias when small IID bias without bounded difference between labeling functions). *Under Assumption 1,*

$$\mathbb{E}_{(x,y) \sim p_T}[\text{bias}^2(x,y)] \leq 2 \times \textit{Correlation shift} + \textit{Support mismatch} + O(\epsilon^2) \qquad (8)$$

*Proof.* We follow the same proof as in Proposition 4, except that we now use: $(a+b)^2 \leq 2(a^2 + b^2)$. Then,

$$\int_{\mathcal{X}_T \cap \mathcal{X}_S} \big(f_T(x) - \bar{f}_S(x)\big)^2 p_T(x)dx = \int_{\mathcal{X}_T \cap \mathcal{X}_S} \big((f_T(x) - f_S(x)) + \big(f_S(x) - \bar{f}_S(x)\big)\big)^2 p_T(x)dx$$

$$\leq 2 \times \int_{\mathcal{X}_T \cap \mathcal{X}_S} (f_T(x) - f_S(x))^2 + \big(f_S(x) - \bar{f}_S(x)\big)^2 p_T(x)dx$$

$$\leq 2 \times \int_{\mathcal{X}_T \cap \mathcal{X}_S} (f_T(x) - f_S(x))^2 p_T(x)dx + 2 \times \int_{\mathcal{X}_T \cap \mathcal{X}_S} \epsilon^2 p_T(x)dx$$

$$\leq 2 \times \int_{\mathcal{X}_T \cap \mathcal{X}_S} (f_T(x) - f_S(x))^2 p_T(x)dx + O(\epsilon^2)$$

$\qquad \square$

## C.4 Variance and diversity shift

We prove the link between variance and diversity shift. Our proof builds upon the similarity between NNs and GPs in the kernel regime, detailed in Appendix C.4.1. We discuss our simplifying Assumption 3 in Appendix C.4.2. We present our final proof in Appendix C.4.3. We discuss the relation between variance and initialization in Appendix C.4.4.

### C.4.1 Neural Networks as Gaussian Processes

We fix $d_S, d_T$ and denote $X_{d_S} = \{x_S\}_{(x_S, y_S) \in d_S}$, $X_{d_T} = \{x_T\}_{(x_T, y_T) \in d_T}$ their respective input supports. We fix the initialization of the network. $l_S$ encapsulates all other sources of randomness.

**Lemma 2** (Inspired from [93]). *Given a NN $f(\cdot, \theta(l_S))$ under Assumption 2, we denote $K$ its neural tangent kernel and $K(X_{d_S}, X_{d_S}) \triangleq (K(x_S, x'_S))_{x_S, x'_S \in X^2_{d_S}} \in \mathbb{R}^{n_S \times n_S}$. Given $x \in \mathcal{X}$, we denote $K(x, X_{d_S}) \triangleq [K(x, x_S)]_{x_S \in X_{d_S}} \in \mathbb{R}^{n_S}$. Then:*

$$\text{var}(x) = K(x, x) - K(x, X_{d_S})K(X_{d_S}, X_{d_S})^{-1}K(x, X_{d_S})^{\intercal}. \qquad (9)$$

*Proof.* Under Assumption 2, NNs are equivalent to GPs. $\text{var}(x)$ is the formula of the variance of the GP posterior given by Eq. (2.26) in [93], when conditioned on $d_S$. This formula thus also applies to the variance $f(\cdot, \theta(l_S))$ when $l_S$ varies (at fixed $d_S$ and initialization). $\qquad \square$

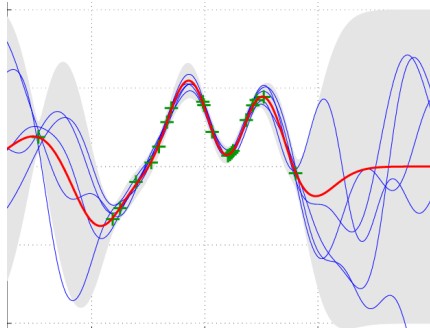

Figure 8: **Mean and variance of a Gaussian process's prediction**. Image from [94]. Intuitively, variance grows when samples are distant from training samples.

### C.4.2 Discussion of the same norm and low similarity Assumption 3 on source dataset

Lemma 2 shows that the variance only depends on the input distributions $p(X)$ without involving the label distributions $p(Y|X)$. This formula highlights that the variance is related to shifts in input similarities (measured by $K$) between $X_{d_S}$ and $X_{d_T}$. Yet, a more refined analysis of the variance requires additional assumptions, in particular to obtain a closed-form expression of $K(X_{d_S}, X_{d_S})^{-1}$. Assumption 3 is useful because then $K(X_{d_S}, X_{d_S})$ is diagonally dominant and can be approximately inverted (see Appendix C.4.3).

The first part of Assumption 3 assumes that $\exists \lambda_S$ such that all training inputs $x_S \in X_{d_S}$ verify $K(x_S, x_S) = \lambda_S$. Note that this equality is standard in some kernel machine algorithms [40, 41, 42] and is usually achieved by replacing $K(x, x')$ by $\lambda_S \frac{K(x,x')}{\sqrt{K(x,x)}\sqrt{K(x',x')}}, \forall (x, x') \in (X_{d_S} \cup X_{d_T})^2$. In the NTK literature, this equality is achieved without changing the kernel by normalizing the samples of $X_{d_S}$ such that they lie on the hypersphere; this input preprocessing was used in [39]. This is theoretically based: for example, the NTK $K(x, x')$ for an architecture with an initial fully connected layer only depends on $\|x\|, \|x'\|, \langle x, x' \rangle$ [95]. Thus in the case where all samples from $X_{d_S}$ are preprocessed to have the same norm, the value of $K(x_S, x_S)$ does not depend on $x_S \in X_{d_S}$; we denote $\lambda_S$ the corresponding value.

The second part of Assumption 3 states that $\exists 0 \leq \epsilon \ll \lambda_S$, s.t. $\forall x_S, x'_S \in X_{d_S}^2, x_S \neq x'_S \Rightarrow |K(x_S, x'_S)| \leq \epsilon$, i.e., that training samples are dissimilar and do not interact. This diagonal structure of the NTK [37], with diagonal values larger than non-diagonal ones, is consistent with empirical observations from [44] at initialization. Theoretically, this is reasonable if $K$ is close to the RBF kernel $K_h(x, x') = \exp(-\|x - x'\|_2^2/h)$ where $h$ would be the bandwidth: in this case, Assumption 3 is satisfied when training inputs are distant in pixel space.

We now provide an analysis of the variance where the diagonal assumption is relaxed. Specifically, we provide the sketch for proving an upper-bound of the variance when the NTK has a block-diagonal structure. This is indeed closer to the empirical observations in [44] at the end of training, consistently with the local elasticity property of NNs [43]. We then consider the dataset $d_{S'} \subset d_S$ made of one sample per block, to which Assumption 3 applies. As decreasing the size of a training dataset empirically reduces variance [96], the variance of $f$ trained on $d_S$ is upper-bounded by the variance of $f$ trained on $d_{S'}$; the latter is given by applying Proposition 3 to $d_{S'}$. We believe that the proper formulation of this idea is beyond the scope of this article and best left for future theoretical work.

### C.4.3 Expression of OOD variance

**Proposition** (3). *Given $f$ trained on source dataset $d_S$ (of size $n_S$) with NTK $K$, under Assumptions 2 and 3, the variance on dataset $d_T$ is:*

$$\mathbb{E}_{x_T \in X_{d_T}}[\text{var}(x_T)] = \frac{n_S}{2\lambda_S}MMD^2(X_{d_S}, X_{d_T}) + \lambda_T - \frac{n_S}{2\lambda_S}\beta_T + O(\epsilon), \qquad (4)$$

*with MMD the empirical Maximum Mean Discrepancy in the RKHS of $K^2(x, y) = (K(x, y))^2$; $\lambda_T \triangleq \mathbb{E}_{x_T \in X_{d_T}} K(x_T, x_T)$ and $\beta_T \triangleq \mathbb{E}_{(x_T, x'_T) \in X_{d_T}^2, x_T \neq x'_T} K^2(x_T, x'_T)$ the empirical mean similarities resp. measured between identical (w.r.t. $K$) and different (w.r.t. $K^2$) samples averaged over $X_{d_T}$.*

*Proof.* Our proof is original and is based on the posterior form of GPs in Lemma 2. Given $d_S$, we recall Equation (9) that states $\forall x \in \mathcal{X}$:

$$\text{var}(x) = K(x,x) - K(x,X_{d_S})K(X_{d_S},X_{d_S})^{-1}K(x,X_{d_S})^{\mathsf{T}}.$$

Denoting $B = K(X_{d_S},X_{d_S})^{-1}$ with symmetric coefficients $b_{i,j} = b_{j,i}$, then

$$\text{var}(x) = K(x,x) - \sum_{\substack{1\leq i\leq n_S \\ 1\leq j\leq n_S}} b_{i,j}K(x,x_S^i)K(x,x_S^j). \tag{10}$$

Assumption 3 states that $K(X_{d_S},X_{d_S}) = A + H$ where $A = \lambda_S \mathbb{I}_{n_S}$ and $H = (h_{ij})_{\substack{1\leq i\leq n_S \\ 1\leq j\leq n_S}}$ with $h_{i,i} = 0$ and $\max_{i,j}|h_{i,j}| \leq \epsilon$.

We fix $x_T \in X_{d_T}$ and determine the form of $B^{-1}$ in two cases: $\epsilon = 0$ and $\epsilon \neq 0$.

**Case when $\epsilon = 0$**   We first derive a simplified result, when $\epsilon = 0$.

Then, $b_{i,i} = \frac{1}{\lambda_S}$ and $b_{i,j} = 0$ s.t.

$$\text{var}(x_T) = K(x_T,x_T) - \sum_{x_S \in X_{d_S}} \frac{K(x_T,x_S)^2}{\lambda_S} = K(x,x) - \frac{n_S}{\lambda_S}\mathbb{E}_{x_S\in X_{d_S}}[K^2(x,x_S)]$$

We can then write:

$$\mathbb{E}_{x_T\in X_{d_T}}[\text{var}(x_T)] = \mathbb{E}_{x_T\in X_{d_T}}[K(x_T,x_T)] - \frac{n_S}{\lambda_S}\mathbb{E}_{x_T\in X_{d_T}}[\mathbb{E}_{x_S\in X_{d_S}}[K^2(x_T,x_S)]]$$

$$\mathbb{E}_{x_T\in X_{d_T}}[\text{var}(x_T)] = \lambda_T - \frac{n_S}{\lambda_S}\mathbb{E}_{x_S\in X_{d_S},x_T\in X_{d_T}}[K^2(x_T,x_S)].$$

We now relate the second term on the r.h.s. to a MMD distance. As $K$ is a kernel, $K^2$ is a kernel and its MMD between $X_{d_S}$ and $X_{d_T}$ is per [97]:

$$\text{MMD}^2(X_{d_S},X_{d_T}) = \mathbb{E}_{x_S\neq x_S'\in X_{d_S}^2}[K^2(x_S,x_S')] + \mathbb{E}_{x_T\neq x_T'\in X_{d_T}^2}[K^2(x_T,x_T')]$$
$$- 2\mathbb{E}_{x_S\in X_{d_S},x_T\in X_{d_T}}[K^2(x_T,x_S)].$$

Finally, because $\epsilon = 0$, $\mathbb{E}_{x_S\neq x_S'\in X_{d_S}^2}K^2(x_S,x_S') = 0$ s.t.

$$\mathbb{E}_{x_T\in X_{d_T}}[\text{var}(x_T)] = \frac{n_S}{2\lambda_S}\text{MMD}^2(X_{d_S},X_{d_T}) + \lambda_T$$
$$- \frac{n_S}{2\lambda_S}\left(\mathbb{E}_{x_T\neq x_T'\in X_{d_T}^2}K^2(x_T,x_T') + \mathbb{E}_{x_S\neq x_S'\in X_{d_S}^2}K^2(x_S,x_S')\right)$$
$$= \frac{n_S}{2\lambda_S}\text{MMD}^2(X_{d_S},X_{d_T}) + \lambda_T - \frac{n_S}{2\lambda_S}\mathbb{E}_{x_T\neq x_T'\in X_{d_T}^2}K^2(x_T,x_T')$$
$$= \frac{n_S}{2\lambda_S}\text{MMD}^2(X_{d_S},X_{d_T}) + \lambda_T - \frac{n_S}{2\lambda_S}\beta_T.$$

We recover the same expression with a $O(\epsilon)$ in the general setting where $\epsilon \neq 0$.

**Case when $\epsilon \neq 0$**   We denote $I : \begin{cases} \text{GL}_{n_S}(\mathbb{R}) & \to \text{GL}_{n_S}(\mathbb{R}) \\ A & \mapsto A^{-1} \end{cases}$ the inversion function defined on $\text{GL}_{n_S}(\mathbb{R})$, the set of invertible matrices of $\mathcal{M}_{n_S}(\mathbb{R})$.

The function $I$ is differentiable [98] in all $A \in \text{GL}_{n_S}(\mathbb{R})$ with its differentiate given by the linear application $dI_A : \begin{cases} \mathcal{M}_{n_S}(\mathbb{R}) & \to \mathcal{M}_{n_S}(\mathbb{R}) \\ H & \mapsto -A^{-1}HA^{-1} \end{cases}$. Therefore, we can perform a Taylor expansion of $I$ at the first order at $A$:

$$I(A+H) = I(A) + dI_A(H) + o(\|H\|),$$
$$(A+H)^{-1} = A^{-1} - A^{-1}HA^{-1} + o(\|H\|).$$

where $\|H\| \le n_S \epsilon = O(\epsilon)$. Thus,

$$(\lambda_S \mathbb{I}_{n_S} + H)^{-1} = (\lambda_S \mathbb{I}_{n_S})^{-1} - (\lambda_S \mathbb{I}_{n_S})^{-1} H (\lambda_S \mathbb{I}_{n_S})^{-1} + O(\epsilon) = \frac{1}{\lambda_S} \mathbb{I}_{n_S} - \frac{1}{\lambda_S^2} H + O(\epsilon),$$

$$\forall i \in [\![1, n_S]\!], b_{ii} = \frac{1}{\lambda_S} - \frac{1}{\lambda_S^2} h_{i,i} + o(\epsilon) = \frac{1}{\lambda_S} + O(\epsilon),$$

$$\forall i \ne j \in [\![1, n_S]\!], b_{ij} = -\frac{1}{\lambda_S^2} h_{i,j} + o(\epsilon) = O(\epsilon).$$

Therefore, when $\epsilon$ is small, Equation (10) can be developed into:

$$\mathrm{var}(x_T) = K(x_T, x_T) - \sum_{x_S \in X_{d_S}} (\frac{1}{\lambda_S} + O(\epsilon)) K(x_T, x_S)^2 + O(\epsilon)$$

$$= K(x_T, x_T) - \frac{n_S}{\lambda_S} \mathbb{E}_{x_S \in X_{d_S}}[K(x_T, x_S)^2] + O(\epsilon)$$

Following the derivation for the case $\epsilon = 0$, and remarking that under Assumption 3 we have $\mathbb{E}_{x_S \ne x'_S \in X_{d_S}^2} K^2(x_S, x'_S) = O(\epsilon^2)$, yields:

$$\mathbb{E}_{x_T \in X_{d_T}}[\mathrm{var}(x_T)] = \frac{n_S}{2\lambda_S} \mathrm{MMD}^2(X_{d_S}, X_{d_T}) + \lambda_T - \frac{n_S}{2\lambda_S} \beta_T + O(\epsilon).$$

$\square$

### C.4.4  Variance and initialization

The MMD depends on the kernel $K$, i.e., only on the initialization of $f$ in the kernel regime per [37]. Thus, to reduce variance, we could act on the initialization to match $p_S(X)$ and $p_T(X)$ in the RKHS of $K^2$. This is consistent with Section 2.4.1 that motivated matching the train and test in features. In our paper, we used the standard pretraining from ImageNet [48], as commonly done on DomainBed [12]. The Linear Probing [49] initialization of the classifier was shown in [49] to prevent the distortion of the features along the training. This could be improved by pretraining the encoder on a task with fewer domain-specific information, e.g., CLIP [99] image-to-text translation as in [36].

### C.5  WA vs. its members

We validate that WA's expected error is smaller than its members' error under the locality constraint.

**Lemma 3** (WA vs. its members.).

$$\mathbb{E}_{L_S^M} \mathcal{E}_T(\theta_{WA}(L_S^M)) - \mathbb{E}_{l_S} \mathcal{E}_T(\theta(l_S)) = \frac{M-1}{M} \mathbb{E}_{x \sim p_T}[\mathrm{cov}(x) - \mathrm{var}(x)] + O(\bar{\Delta}^2) \le O(\bar{\Delta}^2). \quad (11)$$

*Proof.* The proof builds upon Equation (BVCL):

$$\mathbb{E}_{L_S^M} \mathcal{E}_T(\theta_{WA}) = \mathbb{E}_{(x,y) \sim p_T}\Big[\mathrm{bias}(x,y)^2 + \frac{1}{M} \mathrm{var}(x) + \frac{M-1}{M} \mathrm{cov}(x)\Big] + O(\bar{\Delta}^2),$$

and the expression of the standard bias-variance decomposition in Equation (BV) from [32],

$$\mathbb{E}_{l_S} \mathcal{E}_T(\theta) = \mathbb{E}_{(x,y) \sim p_T}\Big[\mathrm{bias}(x,y)^2 + \mathrm{var}(x)\Big].$$

The difference between the two provides:

$$\mathbb{E}_{L_S^M} \mathcal{E}_T(\theta_{WA}) - \mathbb{E}_{l_S} \mathcal{E}_T(\theta) = \frac{M-1}{M} \mathbb{E}_{(x,y) \sim p_T}\Big[\mathrm{cov}(x) - \mathrm{var}(x)\Big] + O(\bar{\Delta}^2).$$

Cauchy Schwartz inequality states $|\mathrm{cov}(Y, Y')| \le \sqrt{\mathrm{var}(Y)\mathrm{var}(Y')}$, thus $\mathrm{cov}(x) \le \mathrm{var}(x)$. Then:

$$\mathbb{E}_{L_S^M} \mathcal{E}_T(\theta_{WA}) - \mathbb{E}_{l_S} \mathcal{E}_T(\theta) \le O(\bar{\Delta}^2).$$

$\square$

## D   Weight averaging versus functional ensembling

We further compare the following two methods to combine $M$ weights $\{\theta(l_S^{(m)})\}_{m=1}^M$: $f_{\text{WA}}$ that averages the weights and $f_{\text{ENS}}$ [15] that averages the predictions. We showed in Lemma 1 that $f_{\text{WA}} \approx f_{\text{ENS}}$ when $\max_{m=1}^M \|\theta(l_S^{(m)}) - \theta_{\text{WA}}\|_2$ is small.

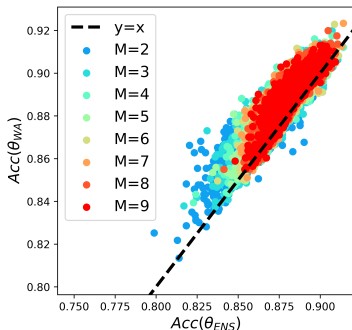

In particular, when $\{l_S^{(m)}\}_{m=1}^M$ share the same initialization and the hyperparameters are sampled from mild ranges, we empirically validate our approximation on OfficeHome in Figure 1. This is confirmed on PACS dataset in Figure 9. For both datasets, we even observe that $f_{\text{WA}}$ performs slightly but consistently better than $f_{\text{ENS}}$. The observed improvement is non-trivial; we refer to Equation 1 in [28] for some initial explanations based on the value of OOD Hessian and the confidence of $f_{\text{WA}}$. The complete analysis of this second-order difference is left for future work.

Figure 9: $f_{\text{WA}}$ performs similarly or better than $f_{\text{ENS}}$ on domain "Art" on PACS.

Yet, we do not claim that $f_{\text{WA}}$ is systematically better than $f_{\text{ENS}}$. In Table 5, we show that this is no longer the case when we relax our two constraints, consistently with Figure 5. *First*, when the classifiers' initializations vary, ENS improves thanks to this additional diversity; in contrast, DiWA degrades because weights are no longer averageable. *Second*, when the hyperparameters are sampled from extreme ranges (defined in Table 7), performance drops significantly for DiWA, but much less for ENS. As a side note, the downward trend in this second setup (even for ENS) is due to inadequate hyperparameters that degrade the expected individual performances.

This highlights a limitation of DiWA, which requires weights that satisfy the locality requirement or are at least linearly connectable. In contrast, Deep Ensembles [15] are computationally expensive (and even impractical for large $M$), but can leverage additional sources of diversity. An interesting extension of DiWA for future work would be to consider the functional ensembling of several DiWAs trained from different initializations or even with different network architectures [100]. Thus the Ensemble of Averages (EoA) strategy introduced in [29] is complementary to DiWA and could be extended into an Ensemble of Diverse Averages.

Table 5: **DiWA's vs. ENS's accuracy** ($\%, \uparrow$) on domain "Art" from OfficeHome when varying initialization and hyperparameter ranges. Best on each setting is in **bold**.

| Configuration | | $M = 20$ | | $M = 60$ | |
|---|---|---|---|---|---|
| Shared classifier init | Mild hyperparameter ranges | DiWA | ENS | DiWA | ENS |
| ✓ | ✓ | **67.3** $\pm$ 0.2 | 66.1 $\pm$ 0.1 | **67.7** | 66.5 |
| ✗ | ✓ | 65.0 $\pm$ 0.5 | **67.5** $\pm$ 0.3 | 65.9 | **68.5** |
| ✓ | ✗ | 56.6 $\pm$ 0.9 | **64.3** $\pm$ 0.4 | 59.5 | **64.7** |

## E   Additional diversity analysis

### E.1   On OfficeHome

#### E.1.1   Feature diversity

In Section 4, our diversity-based theoretical findings were empirically validated using the ratio-error [46], a common diversity measure notably used in [73, 72]. In Figure 10, we recover similar conclusions with another diversity measure: the Centered Kernel Alignment Complement (CKAC) [47], also used in [25, 26]. CKAC operates in the feature space and measures to what extent the pairwise similarity matrices (computed on domain $T$) are aligned — where similarity is the dot product between penultimate representations extracted from two different networks.

#### E.1.2   Accuracy gain per unit of diversity

In Figures 2 and 10a, we indicated the slope of the linear regressions relating diversity to accuracy gain at fixed $M$ (between 2 and 9). For example, when $M = 9$ weights are averaged, the accuracy

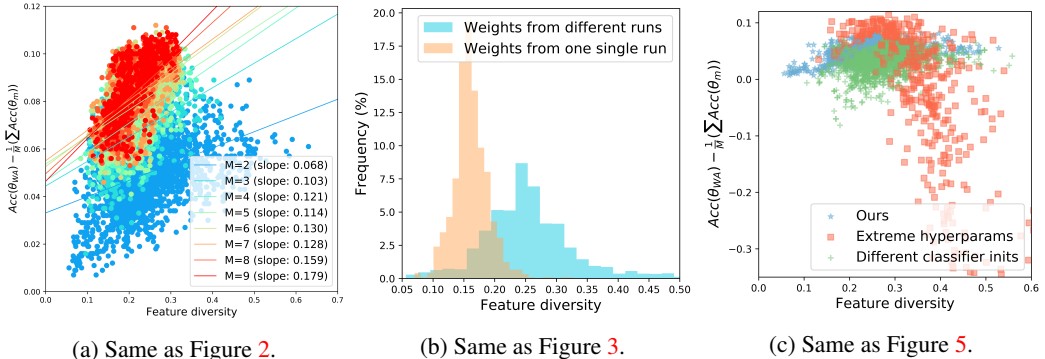

(a) Same as Figure 2.    (b) Same as Figure 3.    (c) Same as Figure 5.

Figure 10: Same analysis as Section 4, where diversity is measured with CKAC [47] in features rather than with ratio-error [46] in predictions.

gain increases by 0.297 per unit of additional diversity in prediction [46] (see Figure 2) and by 0.179 per unit of additional diversity in features [47] (see Figure 10a). Most importantly, we note that the slope increases with $M$. To make this more visible, we plot slopes w.r.t. $M$ in Figure 11. Our observations are consistent with the $(M - 1)/M$ factor in front of $\text{cov}(x)$ in Equation (BVCL). This shows that diversity becomes more important for large $M$. Yet, large $M$ is computationally impractical in standard functional ensembling, as one forward step is required per model. In contrast, WA has a fixed inference time which allows it to consider larger $M$. Increasing $M$ from 20 to 60 is the main reason why DiWA$^{\dagger}$ improves DiWA.

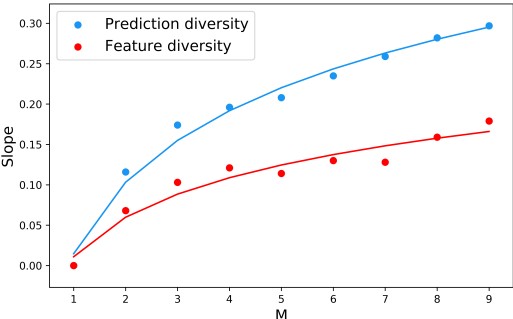

Figure 11: The slopes of linear regression — relating diversity to accuracy gain in Figure 2 and Figure 10a — increases with $M$.

### E.1.3 Diversity comparison across a wide range of methods

Inspired by [21], we further analyze in Figure 12 the diversity between two weights obtained from different (more or less correlated) learning procedures.

- In the upper part, weights are obtained from a single run. They share the same initialization/hyperparameters/data/noise in the optimization procedure and only differ by the number of training steps (which we choose to be a multiple of 50). They are less diverse than the weights in the middle part of Figure 12, that are sampled from two ERM runs.

- When sampled from different runs, the weights become even more diverse when they have more extreme hyperparameter ranges, they do not share the same classifier initialization or they are trained on different data. The first two are impractical for WA, as it breaks the locality requirement (see Figures 5 and 10c). Luckily, the third setting "data diversity" is more convenient and is another reason for the success of DiWA$^{\dagger}$; its 60 weights were trained on 3 different data splits. Data diversity has provable benefits [101], e.g., in bagging [68].

- Finally, we observe that diversity is increased (notably in features) when two runs have different objectives, for example, Interdomain Mixup [56] and Coral [10]. Thus incorporating

weights trained with different invariance-based objectives have two benefits that explain the strong results in Table 2: (1) they learn invariant features by leveraging the domain information and (2) they enrich the diversity of solutions by extracting different features. These solutions can bring their own particularity to WA.

In conclusion, our analysis confirms that "model pairs that diverge more in training methodology display categorically different generalization behavior, producing increasingly uncorrelated errors", as stated in [21].

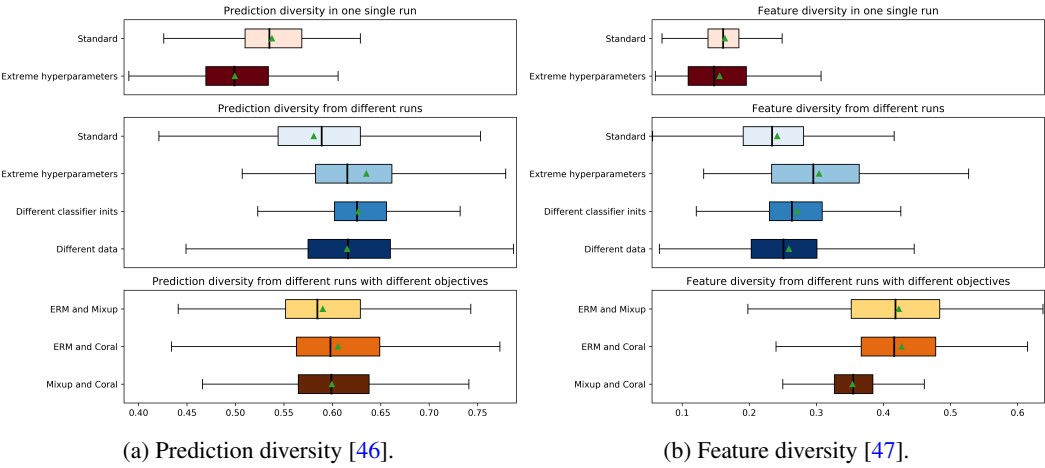

(a) Prediction diversity [46].  (b) Feature diversity [47].

Figure 12: **Diversity analysis** across weights, which are per default trained with ERM, with a mild hyperparameter range (see Table 7), with a shared random classifier initialization, on a given data split. *First*, it confirms Figures 3 and 10b: weights obtained from two different runs are more different than those sampled from a single run (even with extreme hyperparameters). *Second*, this shows that weights from two runs are more diverse when the two runs have different hyperparameters/data/classifier initializations/training objectives. Domain "Art" on OfficeHome.

### E.1.4 Trade-off between diversity and averageability

We argue in Section 2.4.4 that our weights should ideally be diverse functionally while being averageable (despite the nonlinearities in the network). We know from [25] that models fine-tuned from a shared initialization with shared hyperparameters can be connected along a linear path where error remains low; thus, they are averageable as their WA also has a low loss. In Figure 5, we confirmed that averaging models from different initializations performs poorly. Regarding the hyperparameters, Figure 5 shows that hyperparameters can be selected slightly different but not too distant. That is why we chose mild hyperparameter ranges (defined in Table 7) in our main experiments.

A complete analysis of when the averageability holds when varying the different hyperparameters is a promising lead for future work. Still, Figure 13 is a preliminary investigation of the impact of different learning rates (between learning procedures of each weight). First, we validate that more distant learning rates lead to more functional diversity in Figure 13a. Yet, we observe in Figure 13b that if learning rates are too different, weight averaging no longer approximates functional ensembling because the $O(\Delta^2_{L^M_S})$ term in Lemma 1 can be large.

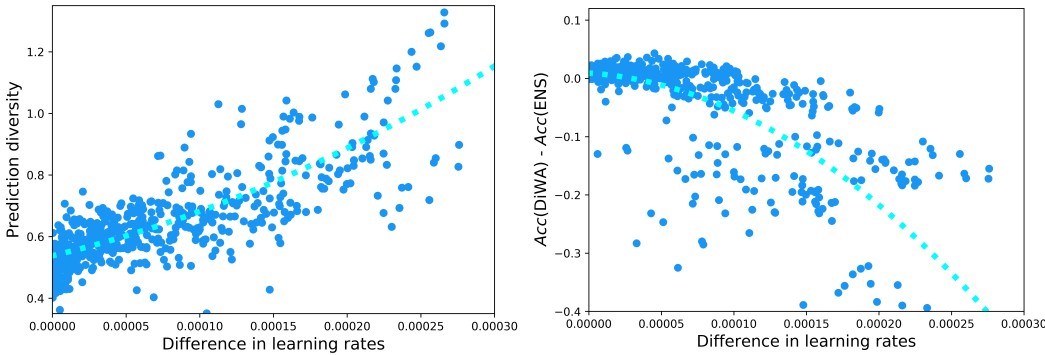

(a) Prediction diversity (↑) [46] between models.

(b) Accuracy (↑) difference between DiWA and ENS.

Figure 13: **Trade-off between diversity and averageability for various differences in learning rates**. Considering $M = 2$ weights obtained from two learning procedures with learning rates $\text{lr}_1$ and $\text{lr}_2$ (sampled from the extreme distribution in Table 7), we plot in Figure 13a the prediction diversity for these $M = 2$ models vs. $|\text{lr}_1 - \text{lr}_2|$. Then, in Figure 13b, we plot the accuracy differences $\text{Acc}(\text{DiWA}) - \text{Acc}(\text{ENS})$ vs. $|\text{lr}_1 - \text{lr}_2|$.

## E.2 On PACS

We perform in Figure 14 on domain "Art" from PACS the same core diversity-based experiments than on OfficeHome in Section 4. We recover the same conclusions.

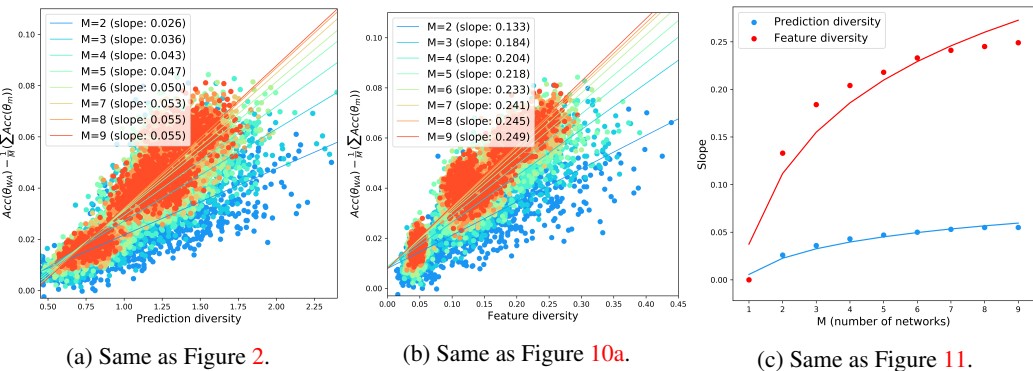

(a) Same as Figure 2.

(b) Same as Figure 10a.

(c) Same as Figure 11.

Figure 14: Same analysis on PACS as previously done on OfficeHome.

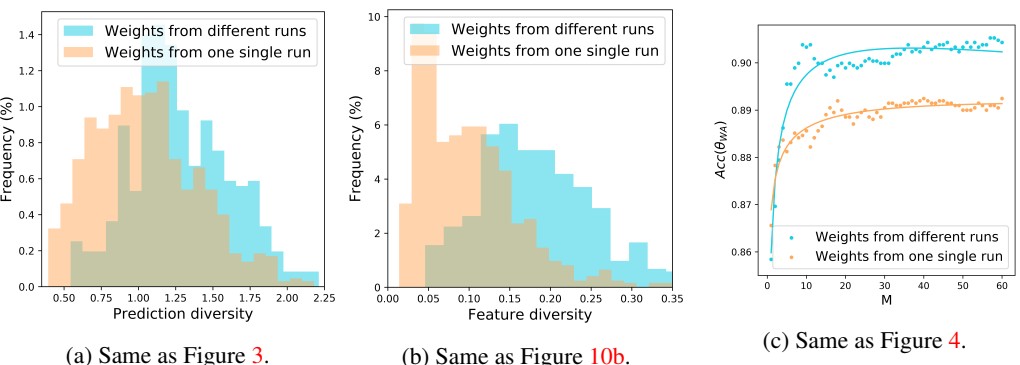

(a) Same as Figure 3.

(b) Same as Figure 10b.

(c) Same as Figure 4.

Figure 15: Same analysis on PACS as previously done on OfficeHome.

# F   Number of training runs

In our experiments, we train 20 independent training runs per data split. We selected this value as 20 is the standard number of hyperparameter trials in DomainBed [12]. In Figure 16 we ablate this choice on the OOD domain "Art" of OfficeHome. We observe that a larger number of runs leads to improved performance and reduced standard deviation. These results are consistent with our theoretical analysis, as the variance is divided per $M$ in Proposition 1. If reducing the training time is critical, one could benefit from significant gains over ERM even with a smaller number of runs: for example, 10 runs seem sufficient in this case. This analysis complements Figure 4 — where 60 runs were launched then sorted in increasing validation accuracy.

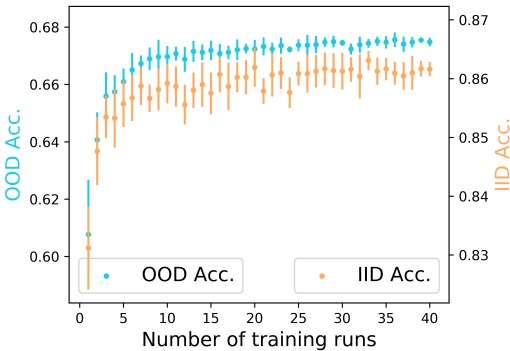

Figure 16: **Mean and standard deviation of DiWA-uniform's accuracy** (↑) on OfficeHome when increasing the number of training runs and uniformly averaging all weights. OOD accuracy is computed on domain "Art", while IID accuracy is computed on validation data from the "Clipart"+"Product"+"Photo" domains.

Moreover, in Table 6 we report DiWA's results when considering only 5 runs, with uniform weight selection. Interestingly, it shows that $M = 5$ is enough to be competitive against SWAD [14], the previous state of the art.

Table 6: **Accuracy** $(\%, \uparrow)$ **on DomainBed**. DiWA-uniform and LP initialization [49].

| Algorithm | PACS | VLCS | OfficeHome | TerraInc | DomainNet | Avg |
|---|---|---|---|---|---|---|
| SWAD [14] | $88.1 \pm 0.1$ | $\mathbf{79.1} \pm 0.1$ | $70.6 \pm 0.2$ | $50.0 \pm 0.3$ | $46.5 \pm 0.1$ | 66.9 |
| DiWA: $M = 5$ | $87.9 \pm 0.2$ | $78.3 \pm 0.3$ | $71.5 \pm 0.2$ | $51.0 \pm 0.7$ | $46.9 \pm 0.3$ | 67.1 |
| DiWA: $M = 20$ | $88.7 \pm 0.2$ | $78.4 \pm 0.2$ | $72.1 \pm 0.2$ | $51.4 \pm 0.6$ | $47.4 \pm 0.2$ | 67.6 |
| DiWA†: $M = 60$ | $\mathbf{89.0}$ | $78.6$ | $\mathbf{72.8}$ | $\mathbf{51.9}$ | $\mathbf{47.7}$ | $\mathbf{68.0}$ |

# G   DomainBed

## G.1   Description of the DomainBed benchmark

We now further detail our experiments on the DomainBed benchmark [12].

**Data.**   DomainBed includes several computer vision classification datasets divided into multiple domains. Each domain is successively considered as the test domain while other domains are used in training. In practice, the data from each domain is split into 80% (used as training and testing) and 20% (used as validation for hyperparameter selection) splits. This random process is repeated with 3 different seeds: the reported numbers are the means and the standard errors over these 3 seeds.

**Training protocol.**   We follow the training protocol from https://github.com/facebookresearch/DomainBed.   For each dataset, domain and seed, we perform a random search of 20 trials on the hyperparameter distributions described in Table 7.   Our mild

distribution is taken directly from [14], yet could be adapted by dataset for better results. Even though these distributions are more restricted than the extreme distributions introduced [12], our ERM runs perform better. It leads to a total amount of 2640 runs only for Table 1. In Appendix B, the $\rho$ hyperparameter for SAM is sampled from $[0.001, 0.002, 0.005, 0.01, 0.02, 0.05]$. In Table 2, hyperparameters specific to Interdomain Mixup [56] ("mixup_alpha") and Coral [10] ("mmd_gamma") are sampled from the distributions defined in [12]. We use a ResNet50 [52] pretrained on ImageNet, with a dropout layer before the newly added dense layer and fine-tuned with frozen batch normalization layers. The optimizer is Adam [102]. Our classifier is either initialized randomly or with Linear Probing [49]; in the latter case, we first learn only the classifier (with the encoder frozen) with the default hyperparameters defined in Table 7; the classifier's weights are then used to initialize all subsequent runs. All runs are trained for 5k steps, except on DomainNet with 15k steps as done in concurrent works [14, 29]. As in [14], validation accuracy is calculated every 50 steps for VLCS, 500 steps for DomainNet and 100 steps for others.

Table 7: Hyperparameters, their default values and distributions for random search.

| | | Random distribution | |
| Hyperparameter | Default value | Extreme (DomainBed [12]) | Mild (DiWA as [14]) |
|---|---|---|---|
| Learning rate | $5 \cdot 10^{-5}$ | $10^{\mathcal{U}(-5,-3.5)}$ | $[1, 3, 5] \cdot 10^{-5}$ |
| Batch size | 32 | $2^{\mathcal{U}(3,5.5)}$ | 32 |
| ResNet dropout | 0 | $[0, 0.1, 0.5]$ | $[0, 0.1, 0.5]$ |
| Weight decay | 0 | $10^{\mathcal{U}(-6,-2)}$ | $[10^{-6}, 10^{-4}]$ |

**Model selection and scores.** We consider the training-domain validation set protocol. From each run, we thus take the weights of the epoch with maximum accuracy on the validation dataset — which follows the training distribution. Our restricted weight selection is also based on this training-domain validation set. This strategy is not possible for DiWA[†] as it averages $M = 20 \times 3$ weights trained with different data splits: they do not share a common validation dataset. The scores for ERM and Coral are taken from DomainBed [12]. Scores for SWAD [14] and MA [29] are taken from their respective papers. Note that MA and SWAD perform similarly even though SWAD introduced three additional hyperparameters tuned per dataset: "an optimum patient parameter, an overfitting patient parameter, and the tolerance rate for searching the start iteration and the end iteration". Thus we reproduced MA [29] which was much easier to implement, and closer to our uniform weight selection.

### G.2 DomainBed results detailed per domain for each real-world dataset

Tables below detail results per domain for the 5 multi-domain real-world datasets from DomainBed: PACS [51], VLCS [53], OfficeHome [50], TerraIncognita [54] and DomainNet [55]. Critically, [19] showed that diversity shift dominates in these datasets.

Table 8: **Accuracy** ($\%, \uparrow$) **on PACS with ResNet50** (best in **bold** and second best underlined).

| | Algorithm | Weight selection | Init | A | C | P | S | Avg |
|---|---|---|---|---|---|---|---|---|
| | ERM | N/A | | $84.7 \pm 0.4$ | $80.8 \pm 0.6$ | $97.2 \pm 0.3$ | $79.3 \pm 1.0$ | $85.5 \pm 0.2$ |
| | Coral[10] | N/A | | $88.3 \pm 0.2$ | $80.0 \pm 0.5$ | $97.5 \pm 0.3$ | $78.8 \pm 1.3$ | $86.2 \pm 0.3$ |
| | SWAD [14] | Overfit-aware | Random | $89.3 \pm 0.5$ | $83.4 \pm 0.6$ | $97.3 \pm 0.3$ | $82.5 \pm 0.8$ | $88.1 \pm 0.1$ |
| | MA [29] | Uniform | | $89.1 \pm 0.1$ | $82.6 \pm 0.2$ | $97.6 \pm 0.0$ | $80.5 \pm 0.9$ | $87.5 \pm 0.2$ |
| | DENS [15, 29] | Uniform: $M=6$ | | $88.3$ | $83.6$ | $96.5$ | $81.9$ | $87.6$ |
| Our runs | ERM | N/A | | $87.6 \pm 0.4$ | $80.1 \pm 1.5$ | $97.7 \pm 0.3$ | $76.7 \pm 1.2$ | $85.5 \pm 0.5$ |
| | MA [29] | Uniform | | $89.9 \pm 0.1$ | $83.3 \pm 0.4$ | $97.8 \pm 0.2$ | $80.6 \pm 0.3$ | $87.9 \pm 0.1$ |
| | ENS | Uniform: $M=20$ | Random | $88.9 \pm 0.4$ | $82.3 \pm 0.5$ | $97.4 \pm 0.3$ | $83.2 \pm 0.3$ | $88.0 \pm 0.1$ |
| | DiWA | Restricted: $M \leq 20$ | | $90.0 \pm 0.3$ | $82.0 \pm 0.5$ | $97.5 \pm 0.1$ | $82.0 \pm 0.6$ | $87.9 \pm 0.2$ |
| | DiWA | Uniform: $M=20$ | | $90.1 \pm 0.6$ | $83.3 \pm 0.6$ | $98.2 \pm 0.1$ | $83.4 \pm 0.4$ | $88.8 \pm 0.4$ |
| | DiWA$^\dagger$ | Uniform: $M=60$ | | $90.5$ | **$83.7$** | $98.2$ | **$83.8$** | **$89.0$** |
| | ERM | N/A | | $86.8 \pm 0.8$ | $80.6 \pm 1.0$ | $97.4 \pm 0.4$ | $78.7 \pm 2.0$ | $85.9 \pm 0.6$ |
| | MA [29] | Uniform | | $89.5 \pm 0.1$ | $82.8 \pm 0.2$ | $97.8 \pm 0.1$ | $80.9 \pm 1.3$ | $87.8 \pm 0.3$ |
| | ENS | Uniform: $M=20$ | | $89.6 \pm 0.2$ | $81.6 \pm 0.3$ | $97.8 \pm 0.2$ | $83.5 \pm 0.5$ | $88.1 \pm 0.3$ |
| | DiWA | Restricted: $M \leq 20$ | LP [49] | $89.3 \pm 0.2$ | $82.8 \pm 0.2$ | $98.0 \pm 0.1$ | $82.0 \pm 0.9$ | $88.0 \pm 0.3$ |
| | DiWA | Uniform: $M=5$ | | $89.9 \pm 0.5$ | $82.3 \pm 0.3$ | $97.7 \pm 0.4$ | $81.7 \pm 0.8$ | $87.9 \pm 0.2$ |
| | DiWA | Uniform: $M=20$ | | $90.1 \pm 0.2$ | $82.8 \pm 0.6$ | **$98.3 \pm 0.1$** | $83.3 \pm 0.4$ | $88.7 \pm 0.2$ |
| | DiWA$^\dagger$ | Uniform: $M=60$ | | **$90.6$** | $83.4$ | $98.2$ | **$83.8$** | **$89.0$** |

Table 9: **Accuracy** ($\%, \uparrow$) **on VLCS with ResNet50** (best in **bold** and second best underlined).

| | Algorithm | Weight selection | Init | C | L | S | V | Avg |
|---|---|---|---|---|---|---|---|---|
| | ERM | N/A | | $97.7 \pm 0.4$ | $64.3 \pm 0.9$ | $73.4 \pm 0.5$ | $74.6 \pm 1.3$ | $77.5 \pm 0.4$ |
| | Coral[10] | N/A | | $98.3 \pm 0.1$ | $66.1 \pm 1.2$ | $73.4 \pm 0.3$ | $77.5 \pm 1.2$ | $78.8 \pm 0.6$ |
| | SWAD [14] | Overfit-aware | Random | $98.8 \pm 0.1$ | $63.3 \pm 0.3$ | $75.3 \pm 0.5$ | $79.2 \pm 0.6$ | $79.1 \pm 0.1$ |
| | MA [29] | Uniform | | **$99.0 \pm 0.2$** | $63.0 \pm 0.2$ | $74.5 \pm 0.3$ | $76.4 \pm 1.1$ | $78.2 \pm 0.2$ |
| | DENS [15, 29] | Uniform: $M=6$ | | $98.7$ | $64.5$ | $72.1$ | $78.9$ | $78.5$ |
| Our runs | ERM | N/A | | $97.9 \pm 0.5$ | $64.2 \pm 0.3$ | $73.5 \pm 0.5$ | $74.9 \pm 1.2$ | $77.6 \pm 0.2$ |
| | MA [29] | Uniform | | $98.5 \pm 0.2$ | $63.5 \pm 0.2$ | $74.4 \pm 0.8$ | $77.3 \pm 0.3$ | $78.4 \pm 0.1$ |
| | ENS | Uniform: $M=20$ | Random | $98.6 \pm 0.1$ | **$64.9 \pm 0.2$** | $73.5 \pm 0.3$ | $77.7 \pm 0.3$ | $78.7 \pm 0.1$ |
| | DiWA | Restricted: $M \leq 20$ | | $98.3 \pm 0.1$ | $63.9 \pm 0.2$ | $75.6 \pm 0.2$ | $79.1 \pm 0.3$ | $79.2 \pm 0.1$ |
| | DiWA | Uniform: $M=20$ | | $98.4 \pm 0.1$ | $63.4 \pm 0.1$ | $75.5 \pm 0.3$ | $78.9 \pm 0.6$ | $79.1 \pm 0.2$ |
| | DiWA$^\dagger$ | Uniform: $M=60$ | | $98.4$ | $63.3$ | **$76.1$** | $79.6$ | **$79.4$** |
| | ERM | N/A | | $98.1 \pm 0.3$ | $64.4 \pm 0.3$ | $72.5 \pm 0.5$ | $77.7 \pm 1.3$ | $78.1 \pm 0.5$ |
| | MA [29] | Uniform | | $98.9 \pm 0.0$ | $62.9 \pm 0.5$ | $73.7 \pm 0.3$ | $78.7 \pm 0.6$ | $78.5 \pm 0.4$ |
| | ENS | Uniform: $M=20$ | | $98.5 \pm 0.1$ | **$64.9 \pm 0.1$** | $73.4 \pm 0.4$ | $77.2 \pm 0.4$ | $78.5 \pm 0.1$ |
| | DiWA | Restricted: $M \leq 20$ | LP [49] | $98.4 \pm 0.0$ | $64.1 \pm 0.2$ | $73.3 \pm 0.4$ | $78.1 \pm 0.8$ | $78.5 \pm 0.1$ |
| | DiWA | Uniform: $M=5$ | | $98.8 \pm 0.0$ | $63.8 \pm 0.5$ | $72.9 \pm 0.2$ | $77.6 \pm 0.5$ | $78.3 \pm 0.3$ |
| | DiWA | Uniform: $M=20$ | | $98.8 \pm 0.1$ | $62.8 \pm 0.2$ | $73.9 \pm 0.3$ | $78.3 \pm 0.1$ | $78.4 \pm 0.2$ |
| | DiWA$^\dagger$ | Uniform: $M=60$ | | $98.9$ | $62.4$ | $73.9$ | $78.9$ | $78.6$ |

Table 10: **Accuracy** ($\%, \uparrow$) **on OfficeHome with ResNet50** (best in **bold** and second best underlined).

| | Algorithm | Weight selection | Init | A | C | P | R | Avg |
|---|---|---|---|---|---|---|---|---|
| | ERM | N/A | | $61.3 \pm 0.7$ | $52.4 \pm 0.3$ | $75.8 \pm 0.1$ | $76.6 \pm 0.3$ | $66.5 \pm 0.3$ |
| | Coral[10] | N/A | | $65.3 \pm 0.4$ | $54.4 \pm 0.5$ | $76.5 \pm 0.1$ | $78.4 \pm 0.5$ | $68.7 \pm 0.3$ |
| | SWAD [14] | Overfit-aware | Random | $66.1 \pm 0.4$ | $57.7 \pm 0.4$ | $78.4 \pm 0.1$ | $80.2 \pm 0.2$ | $70.6 \pm 0.2$ |
| | MA [29] | Uniform | | $66.7 \pm 0.5$ | $57.1 \pm 0.1$ | $78.6 \pm 0.1$ | $80.0 \pm 0.0$ | $70.6 \pm 0.1$ |
| | DENS [15, 29] | Uniform: $M=6$ | | $65.6$ | $58.5$ | $78.7$ | $80.5$ | $70.8$ |
| Our runs | ERM | N/A | | $62.9 \pm 1.3$ | $54.0 \pm 0.2$ | $75.7 \pm 0.9$ | $77.0 \pm 0.8$ | $67.4 \pm 0.6$ |
| | MA [29] | Uniform | | $65.0 \pm 0.2$ | $57.9 \pm 0.3$ | $78.5 \pm 0.1$ | $79.7 \pm 0.1$ | $70.3 \pm 0.1$ |
| | ENS | Uniform: $M=20$ | Random | $66.1 \pm 0.1$ | $57.0 \pm 0.3$ | $79.0 \pm 0.2$ | $80.0 \pm 0.1$ | $70.5 \pm 0.1$ |
| | DiWA | Restricted: $M \leq 20$ | | $66.7 \pm 0.1$ | $57.0 \pm 0.3$ | $78.5 \pm 0.3$ | $79.9 \pm 0.3$ | $70.5 \pm 0.1$ |
| | DiWA | Uniform: $M=20$ | | $67.3 \pm 0.2$ | $57.9 \pm 0.2$ | $79.0 \pm 0.2$ | $79.9 \pm 0.1$ | $71.0 \pm 0.1$ |
| | DiWA$^\dagger$ | Uniform: $M=60$ | | $67.7$ | $58.8$ | $79.4$ | $80.5$ | $71.6$ |
| | ERM | N/A | | $63.9 \pm 1.2$ | $54.8 \pm 0.6$ | $78.7 \pm 0.1$ | $80.4 \pm 0.2$ | $69.4 \pm 0.2$ |
| | MA [29] | Uniform | | $67.4 \pm 0.4$ | $57.3 \pm 0.9$ | $79.7 \pm 0.1$ | $81.7 \pm 0.6$ | $71.5 \pm 0.3$ |
| | ENS | Uniform: $M=20$ | | $67.0 \pm 0.1$ | $57.9 \pm 0.4$ | $80.0 \pm 0.2$ | $81.7 \pm 0.3$ | $71.7 \pm 0.1$ |
| | DiWA | Restricted: $M \leq 20$ | LP [49] | $67.8 \pm 0.5$ | $57.2 \pm 0.5$ | $79.6 \pm 0.1$ | $81.4 \pm 0.4$ | $71.5 \pm 0.2$ |
| | DiWA | Uniform: $M=5$ | | $68.4 \pm 0.4$ | $57.4 \pm 0.5$ | $79.2 \pm 0.2$ | $80.9 \pm 0.4$ | $71.5 \pm 0.3$ |
| | DiWA | Uniform: $M=20$ | | $68.4 \pm 0.2$ | $58.2 \pm 0.5$ | $80.0 \pm 0.1$ | $81.7 \pm 0.3$ | $72.1 \pm 0.2$ |
| | DiWA$^\dagger$ | Uniform: $M=60$ | | **$69.2$** | **$59.0$** | **$80.6$** | **$82.2$** | **$72.8$** |

Table 11: **Accuracy** $(\%, \uparrow)$ **on TerraIncognita with ResNet50** (best in **bold** and second best underlined).

| | Algorithm | Weight selection | Init | L100 | L38 | L43 | L46 | Avg |
|---|---|---|---|---|---|---|---|---|
| | ERM | N/A | | $49.8 \pm 4.4$ | $42.1 \pm 1.4$ | $56.9 \pm 1.8$ | $35.7 \pm 3.9$ | $46.1 \pm 1.8$ |
| | Coral[10] | N/A | Random | $51.6 \pm 2.4$ | $42.2 \pm 1.0$ | $57.0 \pm 1.0$ | $39.8 \pm 2.9$ | $47.6 \pm 1.0$ |
| | SWAD [14] | Overfit-aware | | $55.4 \pm 0.0$ | $44.9 \pm 1.1$ | $59.7 \pm 0.4$ | $39.9 \pm 0.2$ | $50.0 \pm 0.3$ |
| | MA [29] | Uniform | | $54.9 \pm 0.4$ | $45.5 \pm 0.6$ | $60.1 \pm 1.5$ | $40.5 \pm 0.4$ | $50.3 \pm 0.5$ |
| | DENS [15, 29] | Uniform: $M = 6$ | | $53.0$ | $42.6$ | $60.5$ | $40.8$ | $49.2$ |
| Our runs | ERM | N/A | | $56.3 \pm 2.9$ | $43.1 \pm 1.6$ | $57.1 \pm 1.0$ | $36.7 \pm 0.7$ | $48.3 \pm 0.8$ |
| | MA [29] | Uniform | | $53.2 \pm 0.4$ | $46.3 \pm 1.0$ | $60.1 \pm 0.6$ | $40.2 \pm 0.8$ | $49.9 \pm 0.2$ |
| | ENS | Uniform: $M = 20$ | Random | $56.4 \pm 1.5$ | $45.3 \pm 0.4$ | $\mathbf{61.0} \pm 0.3$ | $\underline{41.4} \pm 0.5$ | $51.0 \pm 0.5$ |
| | DiWA | Restricted: $M \le 20$ | | $55.6 \pm 1.5$ | $47.5 \pm 0.5$ | $59.5 \pm 0.5$ | $39.4 \pm 0.2$ | $50.5 \pm 0.5$ |
| | DiWA | Uniform: $M = 20$ | | $52.2 \pm 1.8$ | $46.2 \pm 0.4$ | $59.2 \pm 0.2$ | $37.8 \pm 0.6$ | $48.9 \pm 0.5$ |
| | DiWA† | Uniform: $M = 60$ | | $52.7$ | $46.3$ | $59.0$ | $37.7$ | $49.0$ |
| | ERM | N/A | | $\mathbf{59.9} \pm 4.2$ | $46.9 \pm 0.9$ | $54.6 \pm 0.3$ | $40.1 \pm 2.2$ | $50.4 \pm 1.8$ |
| | MA [29] | Uniform | | $54.6 \pm 1.4$ | $48.6 \pm 0.4$ | $59.9 \pm 0.7$ | $\mathbf{42.7} \pm 0.8$ | $51.4 \pm 0.6$ |
| | ENS | Uniform: $M = 20$ | | $55.6 \pm 1.4$ | $45.4 \pm 0.4$ | $\mathbf{61.0} \pm 0.4$ | $41.3 \pm 0.3$ | $50.8 \pm 0.5$ |
| | DiWA | Restricted: $M \le 20$ | LP [49] | $\underline{58.5} \pm 2.2$ | $48.2 \pm 0.3$ | $58.5 \pm 0.3$ | $41.1 \pm 1.2$ | $\underline{51.6} \pm 0.9$ |
| | DiWA | Uniform: $M = 5$ | | $56.0 \pm 2.5$ | $48.9 \pm 0.8$ | $58.4 \pm 0.2$ | $40.6 \pm 0.8$ | $51.0 \pm 0.7$ |
| | DiWA | Uniform: $M = 20$ | | $56.3 \pm 1.9$ | $\underline{49.4} \pm 0.7$ | $59.9 \pm 0.4$ | $39.8 \pm 0.5$ | $51.4 \pm 0.6$ |
| | DiWA† | Uniform: $M = 60$ | | $57.2$ | $\mathbf{50.1}$ | $60.3$ | $39.8$ | $\mathbf{51.9}$ |

Table 12: **Accuracy** $(\%, \uparrow)$ **on DomainNet with ResNet50** (best in **bold** and second best underlined).

| | Algorithm | Weight selection | Init | clip | info | paint | quick | real | sketch | Avg |
|---|---|---|---|---|---|---|---|---|---|---|
| | ERM | N/A | | $58.1 \pm 0.3$ | $18.8 \pm 0.3$ | $46.7 \pm 0.3$ | $12.2 \pm 0.4$ | $59.6 \pm 0.1$ | $49.8 \pm 0.4$ | $40.9 \pm 0.1$ |
| | Coral[10] | N/A | Random | $59.2 \pm 0.1$ | $19.7 \pm 0.2$ | $46.6 \pm 0.3$ | $13.4 \pm 0.4$ | $59.8 \pm 0.2$ | $50.1 \pm 0.6$ | $41.5 \pm 0.1$ |
| | SWAD [14] | Overfit-aware | | $66.0 \pm 0.1$ | $22.4 \pm 0.3$ | $53.5 \pm 0.1$ | $16.1 \pm 0.2$ | $65.8 \pm 0.4$ | $55.5 \pm 0.3$ | $46.5 \pm 0.1$ |
| | MA [29] | Uniform | | $64.4 \pm 0.3$ | $22.4 \pm 0.2$ | $53.4 \pm 0.3$ | $15.4 \pm 0.1$ | $64.7 \pm 0.2$ | $55.5 \pm 0.1$ | $46.0 \pm 0.1$ |
| | DENS [15, 29] | Uniform: $M = 6$ | | $\mathbf{68.3}$ | $23.1$ | $54.5$ | $\underline{16.3}$ | $66.9$ | $\mathbf{57.0}$ | $\mathbf{47.7}$ |
| Our runs | ERM | N/A | | $62.6 \pm 0.4$ | $21.6 \pm 0.3$ | $50.4 \pm 0.1$ | $13.8 \pm 0.2$ | $63.6 \pm 0.4$ | $52.5 \pm 0.4$ | $44.1 \pm 0.1$ |
| | MA [29] | Uniform | | $64.5 \pm 0.2$ | $22.7 \pm 0.1$ | $53.8 \pm 0.1$ | $15.6 \pm 0.1$ | $66.0 \pm 0.1$ | $55.7 \pm 0.1$ | $46.4 \pm 0.1$ |
| | ENS | Uniform: $M = 20$ | Random | $\underline{67.3} \pm 0.4$ | $22.9 \pm 0.1$ | $54.2 \pm 0.2$ | $15.5 \pm 0.2$ | $67.7 \pm 0.2$ | $\underline{56.7} \pm 0.2$ | $47.4 \pm 0.2$ |
| | DiWA | Restricted: $M \le 20$ | | $65.2 \pm 0.3$ | $23.0 \pm 0.3$ | $54.0 \pm 0.1$ | $15.9 \pm 0.1$ | $66.2 \pm 0.1$ | $55.5 \pm 0.1$ | $46.7 \pm 0.1$ |
| | DiWA | Uniform: $M = 20$ | | $63.4 \pm 0.2$ | $23.1 \pm 0.1$ | $53.9 \pm 0.2$ | $15.4 \pm 0.2$ | $65.5 \pm 0.2$ | $55.1 \pm 0.2$ | $46.1 \pm 0.1$ |
| | DiWA† | Uniform: $M = 60$ | | $63.5$ | $\mathbf{23.3}$ | $54.3$ | $15.6$ | $65.7$ | $55.3$ | $46.3$ |
| | ERM | N/A | | $63.4 \pm 0.2$ | $21.1 \pm 0.4$ | $50.7 \pm 0.3$ | $13.5 \pm 0.4$ | $64.8 \pm 0.4$ | $52.4 \pm 0.1$ | $44.3 \pm 0.2$ |
| | MA [29] | Uniform | | $64.8 \pm 0.1$ | $22.3 \pm 0.0$ | $54.2 \pm 0.1$ | $16.0 \pm 0.1$ | $67.4 \pm 0.0$ | $55.2 \pm 0.1$ | $46.6 \pm 0.0$ |
| | ENS | Uniform: $M = 20$ | | $66.7 \pm 0.4$ | $22.2 \pm 0.1$ | $54.1 \pm 0.2$ | $15.1 \pm 0.2$ | $\underline{68.4} \pm 0.1$ | $55.7 \pm 0.2$ | $47.0 \pm 0.2$ |
| | DiWA | Restricted: $M \le 20$ | LP [49] | $66.7 \pm 0.2$ | $\mathbf{23.3} \pm 0.2$ | $\underline{55.3} \pm 0.1$ | $\underline{16.3} \pm 0.2$ | $68.2 \pm 0.0$ | $56.2 \pm 0.1$ | $\mathbf{47.7} \pm 0.1$ |
| | DiWA | Uniform: $M = 5$ | | $65.7 \pm 0.5$ | $22.6 \pm 0.2$ | $54.4 \pm 0.4$ | $15.5 \pm 0.5$ | $67.7 \pm 0.0$ | $55.5 \pm 0.4$ | $46.9 \pm 0.3$ |
| | DiWA | Uniform: $M = 20$ | | $65.9 \pm 0.4$ | $23.0 \pm 0.2$ | $55.0 \pm 0.3$ | $16.1 \pm 0.2$ | $\underline{68.4} \pm 0.1$ | $55.7 \pm 0.4$ | $47.4 \pm 0.2$ |
| | DiWA† | Uniform: $M = 60$ | | $66.2$ | $\mathbf{23.3}$ | $\mathbf{55.4}$ | $\mathbf{16.5}$ | $\mathbf{68.7}$ | $56.0$ | $\mathbf{47.7}$ |

## H  Failure of WA under correlation shift on ColoredMNIST

Based on Equation (BVCL), we explained that WA is efficient when variance dominates; we showed in Section 2.4.2 that this occurs under diversity shift. This is confirmed by our state-of-the-art results in Table 1 and Appendix G.2 on PACS, OfficeHome, VLCS, TerraIncognita and DomainNet. In contrast, we argue that WA is inefficient when bias dominates, i.e., in the presence of correlation shift (see Section 2.4.1). We verify this failure on the ColoredMNIST [8] dataset, which is dominated by correlation shift [55].

Colored MNIST is a colored variant of the MNIST handwritten digit classification dataset [103] where the correlation strengths between color and label vary across domains. We follow the protocol described in Appendix G.1 except that (1) we used the convolutional neural network architecture introduced in DomainBed [12] for MNIST experiments and (2) we used the test-domain model selection in addition to the train-domain model selection. Indeed, as stated in [19], "it may be improper to apply training-domain validation to datasets dominated by correlation shift since under the influence of spurious correlations, achieving excessively high accuracy in the training environments often leads to low accuracy in novel test environments".

In Tables 13 and 14, we observe that DiWA-uniform and MA both perform poorly compared to ERM. Note that DiWA-restricted does not degrade ERM as it selects only a few models for averaging (low $M$). This confirms that our approach is useful to tackle diversity shift but not correlation shift, for which invariance-based approaches as IRM [8] or Fishr [11] remain state-of-the-art.

Table 13: **Accuracy** ($\%$, $\uparrow$) **on ColoredMNIST**. WA does not improve performance under correlation shift. Random initialization of the classifier. Training-domain model selection.

| | Algorithm | Weight selection | +90% | +80% | -90% | Avg |
|---|---|---|---|---|---|---|
| | ERM | N/A | $71.7 \pm 0.1$ | $72.9 \pm 0.2$ | $10.0 \pm 0.1$ | $51.5 \pm 0.1$ |
| | Coral [10] | N/A | $71.6 \pm 0.3$ | $73.1 \pm 0.1$ | $9.9 \pm 0.1$ | $51.5 \pm 0.1$ |
| | IRM [8] | N/A | $\mathbf{72.5} \pm 0.1$ | $\underline{73.3} \pm 0.5$ | $10.2 \pm 0.3$ | $\mathbf{52.0} \pm 0.1$ |
| | Fishr [11] | N/A | $\underline{72.3} \pm 0.9$ | $\mathbf{73.5} \pm 0.2$ | $10.1 \pm 0.2$ | $\mathbf{52.0} \pm 0.2$ |
| Our runs | ERM | N/A | $71.5 \pm 0.4$ | $73.3 \pm 0.2$ | $\underline{10.3} \pm 0.2$ | $51.7 \pm 0.2$ |
| | MA [29] | Uniform | $68.9 \pm 0.0$ | $71.8 \pm 0.1$ | $10.0 \pm 0.1$ | $50.3 \pm 0.0$ |
| | ENS | Uniform: $M = 20$ | $71.0 \pm 0.2$ | $72.9 \pm 0.2$ | $9.9 \pm 0.0$ | $51.3 \pm 0.1$ |
| | DiWA | Restricted: $M \leq 20$ | $71.3 \pm 0.2$ | $72.9 \pm 0.1$ | $10.0 \pm 0.1$ | $51.4 \pm 0.1$ |
| | DiWA | Uniform: $M = 20$ | $69.1 \pm 0.8$ | $72.6 \pm 0.4$ | $\mathbf{10.6} \pm 0.1$ | $50.8 \pm 0.4$ |
| | DiWA$^\dagger$ | Uniform: $M = 60$ | $69.3$ | $72.3$ | $\underline{10.3}$ | $50.6$ |

Table 14: **Accuracy** ($\%$, $\uparrow$) **on ColoredMNIST**. WA does not improve performance under correlation shift. Random initialization of the classifier. Test-domain model selection.

| | Algorithm | Weight selection | +90% | +80% | -90% | Avg |
|---|---|---|---|---|---|---|
| | ERM | N/A | $71.8 \pm 0.4$ | $72.9 \pm 0.1$ | $28.7 \pm 0.5$ | $57.8 \pm 0.2$ |
| | Coral [10] | N/A | $71.1 \pm 0.2$ | $73.4 \pm 0.2$ | $31.1 \pm 1.6$ | $58.6 \pm 0.5$ |
| | IRM [8] | N/A | $\underline{72.0} \pm 0.1$ | $72.5 \pm 0.3$ | $\underline{58.5} \pm 3.3$ | $\underline{67.7} \pm 1.2$ |
| | Fishr [11] | N/A | $\mathbf{74.1} \pm 0.6$ | $73.3 \pm 0.1$ | $\mathbf{58.9} \pm 3.7$ | $\mathbf{68.8} \pm 1.4$ |
| Our runs | ERM | N/A | $71.5 \pm 0.3$ | $\mathbf{74.1} \pm 0.4$ | $21.5 \pm 1.9$ | $55.7 \pm 0.4$ |
| | MA [29] | Uniform | $68.8 \pm 0.2$ | $72.1 \pm 0.2$ | $10.2 \pm 0.0$ | $50.4 \pm 0.1$ |
| | ENS | Uniform: $M = 20$ | $71.0 \pm 0.2$ | $72.9 \pm 0.2$ | $9.9 \pm 0.0$ | $51.3 \pm 0.1$ |
| | DiWA | Restricted: $M \leq 20$ | $71.9 \pm 0.4$ | $\underline{73.6} \pm 0.2$ | $21.5 \pm 1.9$ | $55.7 \pm 0.8$ |
| | DiWA | Uniform: $M = 20$ | $69.1 \pm 0.8$ | $72.6 \pm 0.4$ | $10.6 \pm 0.1$ | $50.8 \pm 0.4$ |
| | DiWA$^\dagger$ | Uniform: $M = 60$ | $69.3$ | $72.3$ | $10.3$ | $50.6$ |

## I  Last layer retraining when some target data is available

The traditional OOD generalization setup does not provide access to target samples (labelled or unlabelled). The goal is to learn a model able to generalize to any kind of distributions. This is arguably the most challenging generalization setup: under these strict conditions, we showed that

DiWA outperforms other approaches on DomainBed. Yet, in real-world applications, some target data is often available for training; moreover, last layer retraining on these target samples was shown highly efficient in [86, 104]. The complete analysis of DiWA for this new scenario should be properly addressed in future work; yet, we now hint that a DiWA strategy could be helpful.

Specifically, in Table 15, we consider that after a first training phase on the "Clipart", "Product" and "Photo" domains, we eventually have access to some samples from the target "Art" domain (20% or 80% of the whole domain). Following [86], we re-train only the last layer of the network on these samples before testing. We observe improved performance when the (frozen) feature extractor was obtained via DiWA (from the first stage) rather than from ERM. It suggests that features extracted by DiWA are more adapted to last layer retraining/generalization than those of ERM. In conclusion, we believe our DiWA strategy has great potential for many real-world applications, whether some target data is available for training or not.

Table 15: **Accuracy (↑)** on domain "Art" from OfficeHome when some target samples are available for last layer retraining (LLR) [86]. The feature extractor is either pre-trained only on ImageNet (✗), fine-tuned on the source domains "Clipart", "Product" and "Photo" (ERM), or obtained by averaging multiple runs on these source domains (DiWA-uniform $M = 20$).

| Training on source domains | LLR on target domain (% domain in training) | | |
| --- | --- | --- | --- |
| | ✗(0%) | ✓(20%) | ✓(80%) |
| ✗ | - | $61.2 \pm 0.6$ | $74.4 \pm 1.2$ |
| ERM | $62.9 \pm 1.3$ | $68.0 \pm 0.7$ | $74.7 \pm 0.6$ |
| DiWA | $\mathbf{67.3} \pm 0.3$ | $\mathbf{70.4} \pm 0.1$ | $\mathbf{78.1} \pm 0.6$ |