# OpenReview forum: "Diverse Weight Averaging for Out-of-Distribution Generalization"
_NeurIPS.cc/2022/Conference — NeurIPS 2022 Accept_

### Official Review · Reviewer_ySFH · 2022-07-11

**Rating:** 7
**Confidence:** 4
**Soundness:** 3 good
**Presentation:** 3 good
**Contribution:** 3 good

**Summary:**

In this paper, the authors propose an approach to mitigating the generalization problem in computer vision, under the common scenarios of either a difference in the data generating marginals (covariate shift) or a difference in the class-conditional distributions (which they call concept shift or correlation shift).  They connect previous work on the errors incurred by weight averaging and standard ensembling, and offer a new decomposition that better explains when weight averaging fails under correlation shift or covariate shift.  Their proposed method of averaging diverse sets of weights leads to improved results on the DomainBed benchmark suite of datasets.


**Questions:**

One thing I found inconsistent, was the combination of the results of Lemma 1 and the evidence in Figure 1.  Lemma 1 relates the loss of a weight-averaged model and a prediction-averaged model, showing  that the loss of the weight averaged model is the same as the prediction averaged model, plus a (non-negative) term based on the max of the 2-norm between the weight average parameters and an individual member.  Yet if that is so, I would expect prediction averaged models to outperform weight-averaged ones.  Figure 1 establishes that in fact the opposite is true.  Is it the case that I am misinterpreting Lemma 1?

An answer might lie at the end of section 4 on lines 256-266, as well as part of Appendix D: "These experiments confirm that diversity is key as long as the weights remain averageable.".  So, in effect, weight averaging performs well when the members being averaged lie within  a low-error subspace that is compact.  I would like to see this concept of "averagable" explored further.  There seems to be a practical friction between training diverse members of the ensemble (say by varying the initializations) and ensuring that the members do not stray too far apart so as to no longer stay averagable.  A closer investigation of when this happens (say from Figure 5) in terms of violations of the linear mode connectivity would be illuminating.

**Limitations:**

Yes, they have.

**Strengths And Weaknesses:**

### Strengths:

The paper is well written.  The sections are subdivided cleanly and they preview the arguments established in each subsection, culminating in a larger point.  The authors are clearly well versed in the literature of ensembling and transfer learning.

In particular, section 2.4 and its subsections develop the argument for why weight averaged ensembles are robust to covariate shift (or diversity shift) quite convincingly.

Also, the analysis in section 2.2 of why flatness-based minimizers do not have any effect on OOD error was illuminating.

### Weaknesses:

There are a few small issues that I'd like to see addressed:

At the outset in the abstract, the authors state that the best current approaches for out of distribution generalization are derived from averaging the weights along a training run.  This is not quite so, as there are multiple other approaches (SWAG, Deep Ensembles, LNNS, Loss surface simplexes for mode connecting volumes and fast ensembling). These all introduce greater member diversity though different mechanisms, so it would be more correct to broaden the related work section to include them, and to acknowledge them in the abstract.

On lines 185-186 the authors state that hey follow the objective of decorrelating the learning procedures of ensemble members, citing the DICE paper.  It’s an exaggeration to say that the DICE objective is followed in section 3.  The authors of DICE perform considerable work to implement a variational approximation to the conditional entropy bottleneck between two learners.  The authors of this work do not go to such lengths to decorrelate the features extracted from the inputs by ensemble members.

### Comments:

I think the paper would benefit from a perspective that comments upon the difficulty of transfer learning based on the presence of neural collapse, or learning neural network subspaces, especially as the authors note that the size of the space spanned by the furthest member from the barycentre is important.

It's hard to believe that this paper's claims of increased diversity by new initializations are an actual advance, this is present in all the modern ensembling literature (cf. papers descended from the DeepEnsembles literature).  In Table 1,  there does not seem to be a comparison against simple deep ensembles ($f_{ENS}$ in the notation of the paper.)

A question I have with the setup in 2.1 is why would we not want to compare against gains made by fine-tuning in the target distribution.  In most practical scenarios, there is limited target data available to train on.  Certainly in almost all modern NLP tasks, this is a baseline to consider.  Including this baseline, where a model (or ensemble of models) that is trained on S is allowed to be fine-tuned in increments (of data or learning iterations) on data drawn from T before being evaluated, would qualify the degree to which DiWA is a feasible solution to practical transfer learning.

Overall, despite some small quibbles, I find that there is a lot to offer in this paper and the community would benefit from seeing it in a first class venue.

---

> ### Author Response · Authors · 2022-07-31
> **Response to Reviewer ySFH (1/2)**
>
> We thank the reviewer for this rich feedback.
>
> > \#1: *"there are multiple other approaches [...] it would be more correct to broader the related work section to include them, and to acknowledge them in the abstract."*
>
> We followed the reviewer's suggestion and better connected these works in the related work (*Section 6*). Thus, we included this new sentence *l.334-336*: "To induce greater diversity, [Maddox2019] used a high constant learning rate; [Benton2021] explicitly encouraged the weights to encompass more volume in the weight space; [Worstman2021] minimized cosine similarity between weights".
>
> Moreover, we made our abstract less specific (*l.2-5*), by first mentioning ensembling (hinting at [Lakshminarayanan2017]) before introducing weight averaging strategies.
>
> [Maddox2019] A simple baseline for bayesian uncertainty in deep learning. NeurIPS
> [Benton2021] Loss Surface Simplexes for Mode Connecting Volumes and Fast Ensembling. ICML
> [Worstman2021] Learning neural networks subspaces. ICML
> [Lakshminarayanan2017] Simple and scalable predictive uncertainty estimation using deep ensembles. NeurIPS
>
> > \#2: *"Difficulty of transfer learning based on the presence of neural collapse or learning neural subspaces"*
>
> From a loss landscape perspective, previous methods [Maddox2019,Benton2021,Worstman2021] aimed at "explor[ing] the set of possible solutions instead of simply converging to a single point in the weight space" as stated in [Maddox2019]. Similarly, we try to learn diverse and low-error neural subspaces; we do so via several independent training runs from a shared initialization. This is now shortly discussed *l.336-338* in the related work (*Section 6*).
>
> > \#3: *"It’s an exaggeration to say that the DICE objective is followed"*
>
> We deleted this citation in the revision to avoid any confusion. In the initial submission, we wanted to highlight that, while DICE enforced independence via explicit adversarial regularization, DiWA implicitly follows the same goal by decorrelating the learning procedures.
>
> > \#4: *"It's hard to believe that this paper's claims of increased diversity by new initializations are an actual advance [...] There does not seem to be a comparison against simple deep ensembles*
>
> We discussed network initialization (*l.215,l.264,l.929*) to highlight a limitation of DiWA (*l.317*): in contrast to Deep Ensembles [Lakshminarayanan2017], DiWA requires the different training runs to start from a *shared* initialization. DiWA introduces diversity via "different hyperparameters [...], batch orders, data augmentations [...] stochastic noise and number of training steps" (*l.208-210*).
>
> Moreover, we included in the *Table 1* (summarized below) of the revision the scores of a costly Deep Ensembles [Lakshminarayanan2017] baseline (DENS), where the $M=6$ different networks are trained independently from different initializations. Moreover, we also included the scores of an (even more costly) ensembling strategy (ENS) where the $M=20$ weights are those averaged in DiWA (and thus start from a shared initialization). Critically, DiWA$^\dagger$ remains state-of-the-art, partly because it can combine more members (higher $M$) at no inference overhead.
>
> *Table 1* in *Section 5.1 p.8*:
> | Algo. | Init | PACS | VLCS | OH | TI | DN | Avg |
> |---|---|---|---|---|---|---|---|
> | DENS $M=6$ | Random |87.6           | 78.5                       | 70.8                       | 49.2                       | **47.7**           | 66.8|
> | ENS $M=20$ | Random | 88.0±0.1 | 78.7±0.1 | 70.5±0.1 | 51.0±0.5 | 47.4±0.2 | 67.1 |
> | DiWA $M=20$ | Random | 88.8±0.4 | 79.1±0.2 | 71.0±0.1 | 48.9±0.5 | 46.1±0.1 | 66.8 |
> | DiWA$^\dagger$ M=60 | Random | **89.0** | **79.4** | 71.6 | 49.0 | 46.3 | 67.1 |
> | ENS $M=20$| LP | 88.1±0.3 | 78.5±0.1 | 71.7±0.1 | 50.8±0.5 | 47.0±0.2 | 67.2 |
> | DiWA $M=20$ | LP | 88.7±0.2 | 78.4±0.2 | 72.1±0.2 | 51.4±0.6 | 47.4±0.2 | 67.6 |
> | DiWA$^\dagger$ $M=60$ | LP | **89.0** | 78.6 | **72.8** | **51.9** | **47.7** | **68.0** |

---

> > ### Author Response · Authors · 2022-07-31
> > **Response to Reviewer ySFH (2/2)**
> >
> > > \#5: *"fine-tuning in the target distribution"*
> >
> > In the paper, we follow the traditional OOD generalization setup which does not provide access to target samples (even unlabelled). The goal is to learn a model able to generalize to any kind of distribution. This is arguably the most challenging generalization setup: under these strict conditions, we showed that DiWA outperforms other approaches on DomainBed.
> >
> > Yet, we acknowledge that, sometimes, for real-world usages, “there is limited target data available to train on”. Moreover, last layer retraining on these target samples was shown highly efficient in [Kirichenko2022,Rosenfeld2022]: "[training] only for the linear classifier rivals [fine-tuning] the whole network" [Rosenfeld2022]. Thus we included a small experiment in *Table 15* from *Appendix I* (summarized below) highlighting the potential of DiWA in this scenario.
> >
> > On OfficeHome, we consider that after a first training phase on the source “Clipart+Product+Photo” domains, we have access to some samples from the target domain “Art” (20% of the full dataset in the table below). As [Kirichenko2022], we then re-train only the final layer on these target samples before testing. We observe improved performance when the (frozen) feature extractor was obtained (from the first stage) via DiWA rather than from ERM. It suggests that features extracted by DiWA are more adapted to last layer retraining/generalization than those of ERM. In conclusion, we believe DiWA has great potential for many real-world applications, whether some target data is available for training or not.
> >
> > *Table 15* from *Appendix I p.36*:
> > |  Feature extractor training method       | Acc. without target data | Acc. with last layer retraining on 20% of target domain |
> > |----------|----------|----------|
> > | ERM |62.9±1.3|68.0±0.7|
> > | DiWA|**67.3**±0.3|**70.4**±0.1|
> >
> > [Kirichenko2022] Last Layer Re-Training is Sufficient for Robustness to Spurious Correlations. ICMLW SCIS
> > [Rosenfeld2022] Domain-Adjusted Regression or: ERM May Already Learn Features Sufficient for Out-of-Distribution Generalization.
> >
> > > \#6: *Relation between Lemma 1 and *Figure 1**
> >
> > *Lemma 1* shows “that the loss of the weight averaged model is the same as the prediction averaged model“ plus **a big $O$ of** “a (non-negative) term”. $f(x)=g(x)+O(h(x))$ i.i.f. exists $K$ such that $|f(x)-g(x)| < K \times h(x)$ as $x$&rarr;$0$: thus, a big $O$ considers absolute value and does not "preserve the sign". That is why the difference between the two losses in *Lemma 1* can be positive or negative: we only know that its norm is upper-bounded by a factor of $O(\Delta_{L_S^M}^2)$. Empirically in *Figure 1*, this difference seems to be negative as WA performs slightly better than ENS. More generally, as noticed by R.ySFH, whether this difference is positive or negative is related to the averageability of the weights: we detail this fact in the next response.
> >
> > > \#7: *Locality-diversity trade-off*
> >
> > Indeed, our weights should be functionally diverse and *averageable* (despite the nonlinearities in the network). As stated *l.190* in *Section 2.4.4*, "to reduce WA's error in OOD, we thus seek a good trade-off between" those two contradictory requirements. We know from [Neyshabur2020] that models fine-tuned from a shared initialization with shared hyperparameters can be connected along a linear path where error remains low; they are *averageable* as their weight average also has a low loss. In *Figure 5*, we confirmed that averaging models from different initializations performs poorly. Regarding the hyperparameters, *Figure 5*  shows that they can be selected slightly differently but not too distant. That is why we chose mild hyperparameter ranges (defined in *Table 6*) for our main experiments.
> >
> > A complete analysis of when the averageability holds when varying the different hyperparameters is a promising lead for future work. We provide in the newly-added *Figure 14* from *Appendix E.1.4* some preliminary investigation of the impact of different learning rates. First, we validate that more distant learning rates lead to more functional diversity. Second, we observe that $Acc(\text{DiWA}) << Acc(\text{ENS})$ if learning rates are too distant between runs: DiWA no longer approximates ENS because the $O(\Delta_{L_S^M}^2)$ term in *Lemma 1* can be large.
> >
> > [Neyshabur2020] What is being transferred in transfer learning? NeurIPS

---

> > ### Comment · Reviewer_ySFH · 2022-08-08
> > **Thanks for your thoughful responses**
> >
> > I believe your revisions clarify the connections to prior work and remove some confusing claims will better situate this work in the literature, and present a more fulsome picture of the benefits of DiWA.
> >
> > I remain, however, skeptical of the demonstrated benefits of DiWA in Table 1.   The results for the `Random` init comparison, which is the most controlled experiment between `DiWA` and `DENS` shows identical average performance across tasks.  Also `ENS M=20` shows identical average performance with `DiWA M=60`.  Maybe DiWA reduces the variance somewhat between these, but it's hard to know as you don't report distribution information for `DENS`.
> >
> > Only when `DiWA` is given the benefit of an additional technique for transferring class-conditional probabilities with `LP` does it begin to show some benefit over `ENS M=20`.  So I would say a more direct reading of Table 1 is that DiWA enhances the performance benefit of applying LP over the other ensembling approaches.
> >
> > This does not change my assessment of the value of the paper, which I still view as a clear accept.  I hope it's well received in the conference.

---

> > > ### Author Response · Authors · 2022-08-08
> > > **Thank you for your comment**
> > >
> > > We thank you for your positive and encouraging assessment of our paper and are glad that our response addressed your questions.
> > >
> > > As a final note, we do not claim that DiWA improves ENS: rather, our theoretical section leveraged their similarities. A major advantage for DiWA is that it removes the prohibitive inference cost from ENS. A thorough study of when DiWA improves ENS is interesting and best left for future work; apart from initialization and hyperparameters (for which we provided a preliminary study in *Appendix D*), it could take into consideration other factors such as the network's architecture.

---

### Official Review · Reviewer_xhCM · 2022-07-12

**Rating:** 7
**Confidence:** 3
**Soundness:** 4 excellent
**Presentation:** 3 good
**Contribution:** 3 good

**Summary:**

This paper proposes Diverse Weight Averaging (DiWA), which averages weights obtained from different training runs starting from the same initial parameters and mildly different hyperparameters. The paper decomposes the expected OOD error into (bias, variance, covariance, locality) and claims that weight averaging is most effective when the covariance term dominates, verified empirically. They show improved OOD generalization performance on the DomainBed benchmark.

**Questions:**

What is the performance of ENS using the models obtained by DiWA?

What is the rough training cost of obtaining 60 models for the final results? Were you unable to obtain error bounds in table 1 because of the computation costs?

Minor comments
Line 89: “…why WA outperforms in OOD Sharpness-Aware Minimizer…” I’m unsure whether the grammar here is technically wrong, but I took a while to parse this sentence.

**Limitations:**

To my knowledge, this work has no potential negative societal impact other than what was already present in the existing literature.

**Strengths And Weaknesses:**

Overall, I think this is a solid paper with a conceptually simple method and clean writing.

Over ENS, WA has the benefit of requiring only one feedforward at test time. Both Lemma 1 and figure 1 show that WA is a close approximation to ENS in the training setting of DiWA. In fact, WA seems to perform slightly better than ENS.

DiWA is simple and effective, and the experiments show that it can be combined with previous advances for more benefits: LP initialization, leveraging diversity from ERM/Mixup/Coral algorithms, etc.

---

> ### Author Response · Authors · 2022-07-31
> **Response to Reviewer xhCM**
>
> We thank the reviewer for these insightful comments. We answer the questions below.
>
> > \#1: *"What is the performance of ENS using the models obtained by DiWA?"*
>
> In the revision, we updated *Table 1* to include the performance of the functional ensembling (ENS) of the *same* $M=20$ models which are weight averaged in DiWA. As summarized below, we observe that ENS and DiWA-uniform perform similarly - consistently with our theoretical analysis. More precisely, DiWA is better on OfficeHome and PACS (as suggested by *Figures 1* and *9*), in particular with LP initialization (which favors locality thus averageability). Overall, it validates WA as an effective alternative to functional ensembling with no inference overhead.
>
> *Table 1* in *Section 5.1 p.8*:
> | Algo. | Init | PACS | VLCS | OH | TI | DN | Avg |
> |---|---|---|---|---|---|---|---|
> | ENS | Random | 88.0±0.1 | 78.7±0.1 | 70.5±0.1 | 51.0±0.5 | 47.4±0.2 | 67.1 |
> | DiWA | Random | **88.8**±0.4 | **79.1**±0.2 | 71.0±0.1 | 48.9±0.5 | 46.1±0.1 | 66.8 |
> | ENS | LP | 88.1±0.3 | 78.5±0.1 | 71.7±0.1 | 50.8±0.5 | 47.0±0.2 | 67.2 |
> | DiWA | LP | 88.7±0.2 | 78.4±0.2 | **72.1**±0.2 | **51.4**±0.6 | **47.4**±0.2 | **67.6** |
>
>
> > \#2: *"What is the rough training cost of obtaining 60 models for the final results? Were you unable to obtain error bounds in table 1 because of the computation costs?"*
>
> As detailed in the *Author Checklist p.15 l.610-611*, we trained 2640 models to build *Table 1*; this number corresponds to 2 initializations $\times$ (4 datasets $\times$ 4 domains + 1 dataset $\times$ 6 domains) $\times$ 3 data splits $\times$ 20 trials. These runs last approximately 20000 hours on our internal GPU cluster.
>
> As traditionally done on DomainBed, we obtain error bounds for DiWA by evaluating the model on the $3$ different data splits; we empirically observe that these bounds are smaller than for ERM. Yet, it is true that we do not have bounds for DiWA$^\dagger$ which averages all $3 \times 20$ weights trained on the $3$ data splits; computing similar error bounds for DiWA$^\dagger$ would then triple again the number of runs, and thus the computational cost. As stated by the reviewer and *l.295*, this was impractical for "computational reasons".
>
> Yet, a larger number of models $M$ tends to reduce the standard deviation; this is empirically confirmed in the newly-added *Figure 16* in *Appendix F*, consistently with the $1/M$ factor of the variance term in *Proposition 1*. Thus DiWA$^\dagger$'s standard deviation (where $M=60$) can be roughly upper bounded by DiWA's standard deviation (where $M=20$), which is already smaller than ERM's standard deviation (where $M=1$). In conclusion, WA not only improves performance but also narrows margins of error.
>
> > \#3: *"Minor" grammar*
>
> We thank the reviewer for pointing out the lack of clarity *l.88-89*, which we have now fixed.

---

### Official Review · Reviewer_P5Dq · 2022-07-12

**Rating:** 6
**Confidence:** 2
**Soundness:** 3 good
**Presentation:** 3 good
**Contribution:** 2 fair

**Summary:**

In this paper, the authors proposed a simple method to improve the out-of-domain generalization, i.e., by averaging weights from different training runs rather than one run. Some theoretical analysis is also given for explaining weight averaging (WA) and the proposed method. Experiments on the DomainBed benchmark show the performance of the proposed method.

**Questions:**

1. How to decide the total number of runs? Is the number larger usually mean the final performance is better?
2. Is there any intuition of why using MMD in Proposition 3? How about other divergences?

**Limitations:**

The authors have addressed the limitations of their proposed method.

**Strengths And Weaknesses:**

Strengths:
1. The research question of when WA succeeds and how to improve it, is interesting and relatively less explored.
2. The paper provides theoretical insights for WA and the proposed method.

Weaknesses:
1. The writings of the paper may have some overclaim issues. In the abstract, the authors claim that "for out-of-distribution generalization in computer vision, the best current approach averages the weights along a training run" (line 2-3). But from the introduction, it seems that such a claim is only based on the observation from the DomainBed benchmark (line 22-24). I'm wondering does the observation on one benchmark could represent the whole situation in computer vision?
2. The contribution of this paper is quite minimal. Especially regarding the experimental results, it seems there is no large performance improvement of the proposed method when compared with MA [29] in Table 1; under the random initialization, the performance of the proposed method looks similar to the performance of SWAD [14].
3. The proposed method needs to ensemble weights from different runs (e.g., >=20 runs in the experiments), which may cause unnecessary additional computational cost.

---

> ### Author Response · Authors · 2022-07-31
> **Response to Reviewer P5Dq (1/2)**
>
> We thank the reviewer for this feedback. We hope that our responses below address the expressed concerns.
>
> > \#1: *Overclaim issues in the abstract*
>
> We clarified our claim in the abstract of our new revision. We now factually state that: "weight averaging strategies were shown to perform best on the competitive DomainBed benchmark" (*l.2-5*). Yet, it is worth noting that DomainBed is "the standard public benchmark for OOD generalization right now" (R.b9sv), on which existing methods were thoroughly and arguably fairly compared.
>
> > \#2: *"The contribution of this paper is quite minimal."*
>
> We remind our two main contributions, which we believe are significant:
> * Theoretical contributions. As stated by R.b9sv, our “proposed bias-variance-covariance-locality decomposition (*Proposition 1*) is novel and aids our understanding of weight averaging” and "how to improve it" (R.P5Dq). We explain "why weight averaged ensembles are robust to [...] diversity shit" (R.ySFH) by relating correlation shift to bias (*Proposition 2*) and diversity shift to variance (*Proposition 3*). Our theoretical contributions are "verified empirically" (R.xhCM). They solve existing limitations of prior OOD analysis of WA in a "illuminating" (R.ySFH) way. “It directly motivates their proposed DiWA” (R.b9sv).
> * Empirical contributions. Our models are SoTA on DomainBed, at no inference overhead with a fixed architecture. "Experimental setup is sound and the comparisons with baselines fair" (R.b9sv). Our simplest DiWA model (random initialization + uniform weight selection) is already on par with SWAD. DiWA's performance gain over MA is further increased with the LP initialization as it preserves locality across runs (see *l.223*). Overall, after averaging across datasets, our best approach improves SWAD (the previous SoTA) by 1.1 pts, CORAL (the best invariance approach) by 3.4 pts, and ERM by 4.7 pts. Given the competitiveness of the DomainBed benchmark, we believe that these gains are significant.
>
> Finally, we believe DiWA is all the more interesting for the community as DiWA "can be combined with previous advances for more benefits: LP initialization, leveraging diversity from ERM/Mixup/Coral algorithms, etc" (R.xhCM). For example, we could even perform DiWA of SWAD runs, i.e., average several checkpoints from multiple training runs (rather than one checkpoint per training run, as currently done in DiWA). Our proposed diversity-based theory could also pave the way toward new diversity-enforcing strategies for weight averaging.
>
> > \#3: *"Unecessary additional computational cost. How to decide the total number of runs? Is the number larger usually mean the final performance is better?".*
>
> * *Same computational cost as DomainBed's baselines*. In our experiments, we run $20$ independent trainings per data split: this corresponds to the default number of hyperparameter trials in the standard protocol from DomainBed. "Yet, instead of keeping only one run from the grid-search, DiWA leverages $M$ runs" as stated *l.286*. Thus, on DomainBed, DiWA has the same training computational cost as DomainBed's baselines.
> * *Reducing the computational cost*. Yet, we agree that for real-world applications, such a hyperparameter search may be too costly. Thus we added the new *Appendix F* to ablate the importance of the number of training runs for OOD generalization. First, we highlight in *Figure 16* that the OOD accuracy systematically increases with the number of runs (on OfficeHome): this confirms that a larger number of runs usually means better final performance. Second, the newly-added *Table 6* (summarized below) highlights that a *cheap* DiWA-uniform model with only $M=5$ training runs and LP initialization is already on par with SWAD (on the full DomainBed benchmark).
>
> *Table 6* in *Appendix F p.31*:
> |  Algo. | PACS | VLCS | OH | TI | DN | Avg|
> |----------|----------|----------|------------------------|----------|----------|----------|
> | SWAD | 88.1±0.1 | **79.1**±0.1 | 70.6±0.2 | 50.0±0.3 | 46.5±0.1  | 66.9 |
> | DiWA $M=5$ | 87.9±0.2 | 78.3±0.3  | 71.5±0.2  | 51.0±0.7  | 46.9±0.3 | 67.1 |
> | DiWA $M=20$ | **88.7**±0.2 | 78.4±0.2 | **72.1**±0.2 | **51.4**±0.6 | **47.4**±0.2 | **67.6** |
>
> * *Selecting the *best* number of training runs* is a complex challenge that aims at finding an appropriate trade-off between training cost and performance. Future work could use the [elbow method](https://en.wikipedia.org/wiki/Elbow_method_(clustering)) on the plot of OOD accuracy vs. number of training runs, which we provide in the new *Figure 16* for OfficeHome. In the absence of samples from the target distribution, *Figure 16* suggests that the practitioner could use test samples from the source distribution; indeed, in this case, OOD and IID validation accuracies seem to follow the same trend, consistently with [Miller2021].
>
> [Miller2021]: Accuracy on the Line: On the Strong Correlation Between Out-of-Distribution and In-Distribution Generalization. ICML

---

> > ### Author Response · Authors · 2022-07-31
> > **Response to Reviewer P5Dq (2/2)**
> >
> > > \#4: *"Is there any intuition of why using MMD in Proposition 3? How about other divergences?"*
> >
> > Our goal was to find a simple expression of variance as a function of distribution shift to better understand when WA succeeds. There is no direct way to do so with NNs. In contrast, kernel methods offer a closed-form expression (with a kernel function) of the variance at each sample - as illustrated in *Figure 8* where "variance grows when samples are distant from training samples". Fortunately, NNs can actually be seen as kernel methods in the infinite-width limit; the corresponding kernel is the neural tangent kernel denoted $K$. Specifically, two samples $x_1$ and $x_2$ are close w.r.t. $K$ if a weights variation designed to change the prediction for $x_1$ also impacts the prediction for $x_2$ [Charpiat2020].
> >
> > After a few computations (detailed in our proof in *Appendix C.4.3*), the MMD (w.r.t. $K^2$) between marginal distributions naturally appears. As a reminder, while kernels measure similarities between samples, the MMD is the corresponding distance between distributions. Intuitively, the MMD in our formula measures the distance between domains in the pixel space "from the network's perspective" [Charpiat2020]- using the kernel $K^2$ rather than an arbitrary distance. Future work could try to express the variance as a function of other divergences; yet, it would not change our main result showing that variance is related to shift in marginal distributions.
> >
> > [Charpiat2020] Input Similarity from the Neural Network Perspective. NeurIPS

---

> > > ### Author Response · Authors · 2022-08-08
> > > **End of discussion period**
> > >
> > > We hope that our response has satisfactorily addressed Reviewer P5Dq's questions and concerns, in particular by highlighting our contributions. Otherwise, while discussions are still open, we are happy to provide any additional clarification if needed.

---

> > > ### Comment · Reviewer_P5Dq · 2022-08-10
> > > **Thanks for the rebuttal!**
> > >
> > > After reading the rebuttal and the revised version of the paper, I think my concerns are all addressed properly (especially in the revised abstract and Appendix F). I will raise my score to reflect this. Thank the authors very much for the additional empirical results and the clear explanations for my questions.

---

### Official Review · Reviewer_b9sv · 2022-07-12

**Rating:** 8
**Confidence:** 3
**Soundness:** 4 excellent
**Presentation:** 4 excellent
**Contribution:** 4 excellent

**Summary:**

The paper studies why and when averaging the weights of a model improve out-of-domain generalization by way of a new a bias-variance-covariance-locality decomposition of the expected target domain error. They show how diversity is needed and thus propose Diverse Weight Averaging (DiWA) wherein weights of N independent trained models from shared initialization are averaged together (in an optionally greedy fashion). They show how DiWA outperforms weight averaging across N model checkpoints of a single training run and other baselines on the public DomainBed benchmark.


**Questions:**

I would be curious to know how DiWA and WA would fare when the Sharpness-Aware Minimization (SAM) is used in each training run. Perhaps, the benefits of DiWA or WA would be diminish in the presence of SAM?


**Limitations:**

Yes.
* "extreme hyperparameter ranges lead to weights whose average may perform poorly"
* "diversity is key as long as the weights remain averageable" -- linear connectivity in weight space is needed.


**Strengths And Weaknesses:**

Strengths:
* The paper is very well written -- exposition is clear, well-organized, high quality (free of typos), and each section is well-motivated. For example, "Limitations of the flatness-based analysis" discusses why the current explanation for weight averaging's performance is insufficient and why a different type of analysis (like their 4 term decomposition) is needed.
* The proposed bias-variance-covariance-locality decomposition (Proposition 1) is novel and aids our understanding of weight averaging. Furthermore, it directly motivates their proposed DiWA.
* DiWA is evaluated on DomainBed, which is the standard public benchmark for out-of-domain generalization right now. Experimental setup is sound and the comparisons with baselines fair.

Weaknesses:
* Despite DiWA outperforming weight averaging checkpoints of a single run (WA), it is unlikely to be adopted for some use cases. For example, for large language models, there is no extra cost for doing WA but training the model multiple times as needed by DiWA is not practical or feasible. The extra compute is probably better spent scaling the model size or training for longer.

---

> ### Author Response · Authors · 2022-07-31
> **Response to Reviewer b9sv**
>
> We thank the reviewer for these encouraging comments. We answer the questions below.
>
> > \#1: *Training time issue*
>
> Indeed, DiWA with $M$ models requires at least $M$ independent runs. In the paper, “we restricted ourselves to combining only the runs obtained from the standard ERM grid search from DomainBed” (*l.658*) and thus consider $20$ runs per data split. Yet, we agree that $20$ runs may already be too costly for some applications where training time is critical. Thus, we added *Table 6* (summarized below) where we evaluate a DiWA-uniform model with only $M=5$ independent runs; we observe that this cheaper approach is already competitive against SWAD, the previous state of the art. We hope this will encourage researchers to consider DiWA in their large-scale experiments, especially since DiWA has no inference overhead, unlike standard ensembling. Finally, we stress that DiWA finetunes a shared pre-trained encoder model: in particular for NLP, this pretraining step may represent most of the training computational cost.
>
> *Table 6* in *Appendix F p.31*:
> |  Algo. | PACS | VLCS | OH | TI | DN | Avg|
> |----------|----------|----------|------------------------|----------|----------|----------|
> | SWAD | 88.1±0.1 | **79.1**±0.1 | 70.6±0.2 | 50.0±0.3 | 46.5±0.1  | 66.9 |
> | DiWA $M=5$ | 87.9±0.2 | 78.3±0.3  | 71.5±0.2  | 51.0±0.7  | 46.9±0.3 | 67.1 |
> | DiWA $M=20$ | **88.7**±0.2 | 78.4±0.2 | **72.1**±0.2 | **51.4**±0.6 | **47.4**±0.2 | **67.6** |
>
> > \#2: *"How DiWA and WA would fare when SAM is used in each training run?"*
>
> The complementarity between WA strategies and SAM is interesting and is investigated in the IID setting in the recent [Kaddour2022]. *Appendix B.3* questioned this complementarity in the OOD setting.
> * MA/SWAD + SAM. We actually provided some elements of response in our initial submission. We showed in *Figure 6* that, even if combining MA and SAM further flattens the loss landscape compared to MA alone, "this is not reflected in the OOD accuracies" (*l.700*). This conclusion holds for SWAD + SAM, as reported in *Table 3*.
> * DiWA + SAM. We added the new *Table 4* (summarized below) to show that DiWA-uniform performs slightly worse when SAM is used in each training run (on OfficeHome domain "Art").
>
> *Table 4* in *Appendix B.3 p.18*:
> | Algo.       | with ERM | with SAM |
> |--------| --------   | --------   |
> | No DiWA | 62.9±1.3 | 63.5±0.5|
> | DiWA $M=20$       | 67.3±0.3      | 66.7±0.2      |
> | DiWA$^\dagger$ $M=60$      | **67.7**      | **67.4**      |
>
> These experiments validate that SAM is not complementary with weight averaging strategies for OOD generalization. This (perhaps surprising) phenomenon highlighted the limitation of the previous flatness-based analysis and motivated our diversity-based analysis, which better explains this result: indeed, as shown in *Figure 7*, weights from SAM are less diverse than those from ERM. This lack of diversity is critical for scenarios with diversity shift where variance dominates.
>
> [Kaddour2022] Questions for flat-minima optimization of modern neural networks. arXiv

---

### Author Response · Authors · 2022-07-31
**General Response to All Reviewers**

We sincerely thank the reviewers for their thoughtful remarks and overall positive comments.  We are encouraged that they found our paper "well written" (R.b9sv, R.ySFH) and "solid" (R.xhCM): it "aids our understanding of weight averaging" (R.b9sv) and "how to improve it" (R.P5Dq). The theoretical analysis is "novel" (R.b9sv) and "illuminating" (R.ySFH): it "motivates the proposed DiWA" (R.b9sv), which is "simple" (R.xhCM,R.P5Dq) yet "effective" as it "improved OOD generalization performance" (R.xhCM), with "fair" "comparisons with baselines" (R.b9sv). We answer their questions individually and would be happy to answer any other questions if necessary.

Moreover, following their feedback, we provided a revision of our initial submission (merging the main paper and the supplementary material), where modifications are colored in *navy blue*. In summary,
* in the main paper:
  * we adapted our abstract (*l.2-5* R.P5Dq, R.ySFH), clarified two formulations (*l.88-89* R.xhCM, *l.185-186* R.ySFH) and added two sentences in the related work (*l.334-338* R.ySFH).
  * we updated *Table 1*, adding two costly functional ensembling strategies as new baselines (R.xhCM, R.ySFH).
* in the supplementary material, we added four experiments to answer the reviewers' questions:
  * *Table 4* in *Appendix B.3* p.18: we validate that SAM and DiWA are not complementary for OOD generalization (R.b9sv).
  * *Figure 13* in *Appendix E.1.4* p.29: we show the trade-off between diversity and averageability when varying the learning rates (R.ySFH).
  * *Figure 16* and *Table 6* in *Appendix F* p.31: we analyze how OOD accuracy increases with additional number of training runs (R.b9sv, R.P5Dq).
  * *Table 15* in *Appendix I* p.35: we include a preliminary experiment suggesting that our DiWA strategy is also useful in the scenario where some target data is available for training (R.ySFH).

---

### Meta-Review · Area_Chair_CAzH · 2022-08-26

**Recommendation:** Accept
**Confidence:** Certain

**Metareview:**

The reviewers unanimously recommend accepting the paper - congratulations!

My only concern is in the related work: The submission mentions

> The recent “Model soups” by Wortsman et al. [28] developed a WA algorithm similar to Algorithm 1. However the task, the theoretical analysis and most importantly the goals of these two works are different.

This is not an accurate characterization because Wortsman et al. [28] were also interested in out-of-distribution generalization - their paper mentions "robustness" and "distribution shift" several times and contains results on multiple OOD test sets. The results in this submission and in Wortsman et al. [28] reinforce each other since the two papers evaluate on different OOD benchmarks and find that weight averaging helps in both. I encourage the authors to clarify this in their related work section so that the reader can correctly put the results in context.

**Award:**

No

---

### Decision · Program_Chairs · 2022-09-14

Accept